

# Inferring methane emissions from African livestock by fusing drone, tower, and satellite data

Alouette van Hove[1], Kristoffer Aalstad[1], Vibeke Lind[2], Claudia Arndt[3], Vincent Odongo[3], Rodolfo Ceriani[4,5], Francesco Fava[5], John Hulth[1], and Norbert Pirk[1]

[1]Department of Geosciences, University of Oslo (UiO), Oslo, Norway
[2]Norwegian Institute of Bioeconomy Research (NIBIO), Tjøtta, Norway
[3]Mazingira Centre, International Livestock Research Institute (ILRI), Nairobi, Kenya
[4]Department of Agricultural and Environmental Sciences, University of Milan (UNIMI), Milan, Italy
[5]Department of Environmental Science and Policy, University of Milan (UNIMI), Milan, Italy

**Correspondence:** Alouette van Hove (a.van.hove@geo.uio.no)

**Abstract.**

Considerable uncertainties and unknowns remain in the regional mapping of methane sources, especially in the extensive agricultural areas of Africa. To address this issue, we developed an observing system that estimates methane emission rates by assimilating drone and flux tower observations into an atmospheric dispersion model. In this study, we apply this approach to

verify and quantify potential methane sources identified through radiance anomalies observed in hyperspectral satellite data. We compare different methods to estimate emissions from various ruminant livestock species in sub-Saharan Africa, including diverse herds of cattle, goats, and sheep, as well as camels, for which methane emission estimates are particularly sparse. Our estimates, derived from Bayesian inference, align with Tier 2 emission values of the Intergovernmental Panel on Climate Change. We moreover observe the hypothesized increase in methane emissions following feeding. Our findings suggest that

the Bayesian inference method is more robust under non-stationary wind conditions compared to a mass balance approach using drone observations. Furthermore, the Bayesian inference method performs better in quantifying emissions from weaker sources, estimating methane emission rates as low as $100\,\mathrm{g\,h^{-1}}$. We find a $\pm50\%$ uncertainty in emission rate estimates for these weaker sources, such as sheep and goat herds, which reduces to $\pm12\%$ for stronger sources, like cattle herds emitting $1{,}000 - 1{,}500\,\mathrm{g\,h^{-1}}$. These promising results demonstrate the potential and efficacy of the Bayesian inference method for

source term estimation. Future applications of drone-based Bayesian inference could extend to estimating methane emissions in Africa and other regions from various sources with complex spatiotemporal emission patterns, such as wetlands, landfills, and wastewater disposal sites. The Bayesian observing system could thereby contribute to the validation and improvement of climate models and emission inventories.

## 1 Introduction

While methane ($CH_4$) emissions from so-called super-emitters ($> 100\,\mathrm{kg\,h^{-1}}$) can be quantified using satellite data, the current spatial resolution and spectral resolution, as well as the precision of satellite observations, are not sufficient to effectively



quantify weaker sources, such as livestock herds (Sherwin et al., 2023). To address this limitation and complement the regional mapping of $CH_4$ sources of various strengths, we developed an observing system that uses a Bayesian inference approach to assimilate drone and flux tower observations in an atmospheric dispersion model to estimate $CH_4$ emission rates. We assess

the efficacy of the Bayesian inference method in quantifying $CH_4$ emissions from nine different ruminant herds in sub-Saharan Africa - an important, yet poorly understood, source in the global $CH_4$ emission inventory. By comparing our results to those obtained using a mass balance method with drone observations and to emission values of the Intergovernmental Panel on Climate Change (IPCC) (Gavrilova et al., 2019), we demonstrate the effectiveness and robustness of our approach.

Global mean atmospheric $CH_4$ concentrations surpassed $1.90\,\mathrm{ppm}$ in 2022, marking a $16\,\%$ increase since 1985 (Lan et al.,

2024). Livestock production is a major contributor to global anthropogenic $CH_4$ emissions, accounting for approximately one-third of the total emissions (Saunois et al., 2020). Within this sector, enteric fermentation in ruminants - such as cattle, sheep, goats, and camels - is the predominant source, generating approximately $80\,\%$ of these emissions, while the remaining $20\,\%$ originates from manure (Amon et al., 2001). During the digestive process, $CH_4$ is produced by rumen fermentation, with about $90 - 95\,\%$ released through burping and $5 - 10\,\%$ as intestinal gas (Broucek, 2014). Because $CH_4$ has a high global

warming potential and a relatively short atmospheric lifetime, reducing its emissions can have quick benefits in mitigating climate change (Szopa et al., 2021). Therefore, accurate measurements and understanding of $CH_4$ emissions from ruminants are important for developing effective mitigation strategies and evaluating their efficacy.

The IPCC provides internationally recognized standardized methodologies for estimating $CH_4$ emissions from ruminants (Gavrilova et al., 2019). Tier 1 methods use generalized default values for emission factors, which are often continent or region specific.

In contrast, Tier 2 values incorporate more detailed herd-specific or animal-specific data. These data account for local variations in livestock breeds, manure management practices, feed quality, and environmental conditions. Recent studies from sub-Saharan Africa have demonstrated substantial differences between emission estimates from these two tiers (Goopy et al., 2018; Ndung'u et al., 2019; Gurmu et al., 2024), highlighting the need for precise, locally relevant data. However, there is a scarcity of studies focusing on $CH_4$ emission rates from ruminants in this region. Many studies use energy balance estimates

based on factors such as animal weight, feed, and activity level, with few studies actually measuring $CH_4$ concentrations (as done by e.g., Korir et al., 2022a; Goopy et al., 2020; Mwangi et al., 2023; Wolz et al., 2022). We aim to contribute to help bridge this knowledge gap.

Specifically, research on $CH_4$ emissions from camels is sparse. Although a few studies have estimated emissions using direct $CH_4$ measurements from smaller camelids such as alpacas and llamas (e.g., Pinares-Patiño et al., 2003; Nielsen et al.,

2014), research on larger camelids, like dromedaries and Bactrian camels, remains limited. This lack of data is likely due to the respiration chambers used for measuring gas exchange being too small to fit these larger animals. Nonetheless, Dittmann et al. (2014) conducted respiration chamber measurements with Bactrian camels. Our study represents one of the first efforts to estimate $CH_4$ emissions from camels, specifically dromedaries, thereby contributing to the development of knowledge in a relatively unstudied field.

Gas exchange methods such as respiration chambers and headboxes are typically used to quantify $CH_4$ emissions from individual animals. For estimating emissions from ruminant herds or entire farm facilities, several indirect techniques have been



applied. Tracer-ratio experiments (Vechi et al., 2022; Daube et al., 2019; Arndt et al., 2018) involve releasing a known quantity of a tracer gas and comparing its dispersion to that of $CH_4$. The mass balance approach (Vinković et al., 2022; Arndt et al., 2018; Wratt et al., 2001) calculates emissions based on the difference between incoming and outgoing $CH_4$ flux estimates in a

defined volume. Other studies (Wolz et al., 2022; Bai et al., 2021; Arndt et al., 2018) use open-path Fourier transform infrared or laser spectrometry to obtain horizontal path-integrated $CH_4$ concentrations upwind and downwind from the source. These concentration data are combined with a Lagrangian particle dispersion model to estimate emission rates. Inverse modeling techniques (Andersen et al., 2021) infer emission rates by fitting an atmospheric dispersion model to measured atmospheric data, possibly incorporating prior knowledge.

On much larger spatial scales, satellite observations are frequently used to detect and quantify $CH_4$ emissions from super-emitters, such as leaks from oil and gas production and large landfills (e.g., Pandey et al., 2019; Dogniaux et al., 2024). However, the emission rates of livestock herds are much smaller, and therefore more challenging to identify as potential $CH_4$ sources from satellite data. To accurately map the emission plumes from these weaker sources, measurement platforms with a higher spatial resolution are needed.

The advent of drone technology as a versatile platform for carrying measurement equipment has enabled high-resolution spatiotemporal observations of atmospheric gases and thermodynamic variables (Villa et al., 2016; Burgués and Marco, 2020). This innovation allows for data collection over a larger spatial region than fixed flux towers or monitoring stations and offers higher spatial resolution compared to satellite-based measurements. Moreover, drones facilitate measurements in locations that are otherwise inaccessible to crewed aircrafts due to safety reasons, such as areas close to the $CH_4$ source and near the ground.

In our study, we utilize drones to sample and map $CH_4$ emission plumes from African ruminant herds, leveraging this data to estimate their $CH_4$ production. Notably, this research is, as far as we know, a pioneering effort to employ drones for $CH_4$ emission quantification from ruminants in sub-Saharan Africa. We apply two distinct methods: a traditional mass balance method and an innovative Bayesian inference approach that uses a sequential Monte Carlo method to invert an atmospheric diffusion model. To complement this analysis, we assess the capability of hyperspectral satellite data to pinpoint the location

of $CH_4$ sources, specifically ruminant herds, by identifying spectral anomalies at the landscape level.

While Bayesian inference has previously been applied with drone observations in homogeneous environments with artificial gas sources (Hutchinson et al., 2019, 2020; Park et al., 2021), to the best of our knowledge, this research marks the first application of the method for quantifying ruminant emissions in real-world conditions using drone observations. The Bayesian framework addresses uncertainties in atmospheric observations, potentially enhancing the precision and reliability of emission

estimates. Furthermore, it allows for the integration of observations from multiple platforms, including drones and flux towers. We explore which sensor observations - specifically $CH_4$ concentration measurements and wind measurements, either from the drone or the flux tower - are most effective for applying the Bayesian inference approach.

We aim to achieve the following objectives: (1) To investigate whether spectral indices related to $CH_4$ emissions from hyperspectral satellite data can aid in detecting the locations of $CH_4$ sources, specifically ruminant herds. (2) To evaluate

the efficacy of the Bayesian inference method utilizing drone-based observations for estimating $CH_4$ emission rates. (3) To determine emission rates for free-grazing cattle, sheep, goats, and camels in a sub-Saharan African country using drone-based





observations. (4) To compare the results obtained from the Bayesian inference method with estimates from a mass balance method and IPCC Tier 2 values, to evaluate different methods for estimating $CH_4$ emissions from ruminants in sub-Saharan Africa and to contribute to the improvement of national greenhouse gas inventories.

## 2 Materials and methods

This study was conducted at the Kapiti Research Station in Kenya, approximately $60\,km$ south-east from Nairobi. The station is managed by the International Livestock Research Institute (ILRI). Covering over 13,000 hectares, the station houses various ruminants, including cattle, sheep, goats, and camels, with a primary focus on studying livestock productivity. Livestock management at Kapiti follows typical pastoral systems, where herders allow the animals to graze freely during the day and keep them in enclosures, known in Kenya as bomas, during the night.

This section outlines the different methodologies used in this study to detect and estimate $CH_4$ sources. We apply four approaches: (1) $CH_4$ source location detection through hyperspectral satellite observations, and three methods for source term estimation: (2) $CH_4$ source term estimation through drone observations using a particle-based Bayesian inference method, (3) $CH_4$ source term estimation through drone observations using a mass balance approach, and (4) calculation of IPCC Tier 2 emission values.

### 2.1 Satellite observations for source detection

In an exploratory effort, we investigate the potential for detecting livestock herds as $CH_4$ sources using satellite hyperspectral imagery. On 6 March 2024, the PRecursore IperSpettrale della Missione Applicativa (PRISMA) satellite (Loizzo et al., 2018) was commissioned to capture a hyperspectral image of Kapiti, while three cattle herds were present at different sites. The PRISMA satellite has two hyperspectral sensors that cover a spectral range from $400\,nm$ to $2500\,nm$. Methane exhibits strong absorption features in the Shortwave Infrared (SWIR) region between $2150\,nm$ and $2500\,nm$, with particularly strong absorption around $2300\,nm$ (Moorhead, 1932; Brown et al., 2003; Roger et al., 2024a). Consequently, the Simple Ratio (SR) index of the wavelengths at $2300\,nm$ and $2100\,nm$ (SR2300/2100) is commonly used to detect spatial variations in $CH_4$ absorption (Xiao et al., 2020; Scafutto et al., 2021; Roger et al., 2024b; Pei et al., 2023). A lower SR indicates lower relative radiance at $2300\,nm$ and thus greater absorption, suggesting higher atmospheric $CH_4$ concentrations. However, it is important to note that spatial variations in other factors, such as vegetation water content, leaf structure, and soil moisture, can also influence the SR index.

We process the hyperspectral data of the PRISMA satellite in the infrared region to detect spatial variations in the $CH_4$ absorption feature. Starting from Level-1 top-of-atmosphere radiance narrowbands (Giardino et al., 2020), infrared information is integrated into single data cubes using the PRISMAread package in the R environment (Busetto and Ranghetti, 2020). This produces hyperspectral data cubes consisting of 173 bands, spanning infrared wavelengths from $920\,nm$ to $2505\,nm$. Finally, we calculate the SR index for each data cube using the SWIR wavelengths at $2300\,nm$ and $2100\,nm$.





## 2.2  Drone-based source term estimation

This section provides details of the drone field campaign, conducted between 29 February and 7 March 2024 at Kapiti. It
outlines both the Bayesian inference method and the mass balance approach used to estimate $CH_4$ emission rates from drone
observations.

We use a drone equipped with a gas sensor to obtain $CH_4$ concentration observations of the emission plumes of nine different
ruminant herds: cattle (cows, heifers, steers, and slick herd), sheep (lactating ewes), goats (dry does, pregnant does, weaner
kids), and camels. The cows, heifers, and steers are Boran cattle, while the slick herd is a crossbreed between Holstein-Friesian
and Boran heifers. The sheep flock consists of Red Masaai and Dorper, and the goat herds comprise Small East African and
Galla varieties. The camels are dromedaries. The lactating ewes had lambs, and the pregnant does had kids with them. However,
since the rumen fermentation systems of milk-fed lambs and kids are not yet fully developed (Baldwin et al., 2004), we assume
their $CH_4$ emissions to be negligible and treat these herds as if the lambs and kids were not present. The herd sizes are included
in Table B1.

During the drone flights, the respective herds were confined within a boma at coordinates $-1.61365°N$, $37.13234°E$. The
animals exhibited no signs of distress and appeared at ease throughout the drone operations. Figure 1 shows the heifers inside
the boma during a drone flight that coincided with the satellite overpass, as well as a herd of camels observing a passing drone.

Typically, four flights were conducted for each ruminant herd. In the morning, before grazing, two flights were performed:
one flight for each emission estimation method, namely the Bayesian inference approach and a mass balance approach. The
same set of flights was repeated in the afternoon after the animals had grazed. Feed intake is known to increase enteric $CH_4$
emissions in ruminants, with peak emissions occurring shortly after feeding (Amon et al., 2001; Hegarty, 2013). Since the
animals had no access to feed during the night, lower emissions are expected in the morning compared to the afternoon,
following grazing. We investigate whether there is a noticeable increase in $CH_4$ emissions between the morning and afternoon
flights, using consistent observations of such increases as indicators of the method's reliability and accuracy.

During control drone flights, conducted without animals present in the boma or the immediate surroundings, no increase
in $CH_4$ levels was observed throughout the field campaign. Based on this observation, we assume that $CH_4$ production from
manure is negligible in our study and attribute the elevated $CH_4$ concentrations above the background level solely to enteric
fermentation.

### 2.2.1  Observing system

Our observing system consists of a DJI M300 RTK drone equipped with an AERIS MIRA Strato LDS $CH_4$ gas sensor, as well
as a stationary flux tower with an eddy covariance system (Burba, 2013). The tower is located at coordinates $-1.61419°N$,
$37.13313°E$, approximately $100\,m$ south-south-east from the center of the boma, as shown in Fig. 2.

The drone's position was recorded using Real-Time Kinematic (RTK) positioning. A Digital Elevation Model (DEM) of the
area was obtained using DJI L1 Lidar, processed in DJI Terra. The altitude data of the RTK system was corrected using the



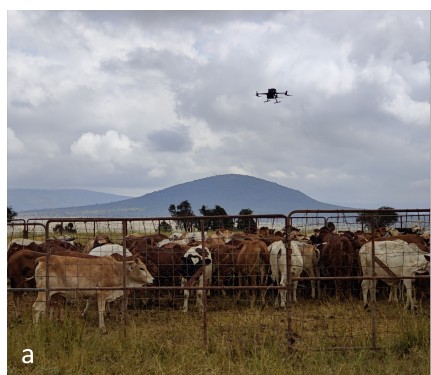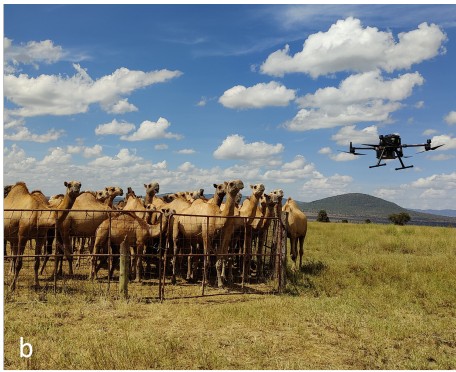

**Figure 1.** (a) A drone flight capturing methane concentration observations of a heifer herd coinciding with the satellite overpass on 6 March 2024. (b) Camels inside the boma during a drone flight on 4 March 2024. Of all animals, the camels were the most curious about the drone.

DEM at the drone's home location. Additionally, the DEM was used to determine the drone's flight height above the ground surface.

The AERIS MIRA Strato LDS gas analyzer detects $CH_4$ using mid-infrared laser spectroscopy, which measures the absorption of infrared radiation by $CH_4$ molecules. The reported mixing ratio $X$ [ppm; parts per million per volume] at the point measurement is the fraction of $CH_4$ molecules per million molecules of air. The sensor has a sensitivity of $1\,\mathrm{ppb}$ and a sampling

rate of $1\,\mathrm{Hz}$. The mixing ratio $X$ is converted to mass concentration $c\,[\mathrm{g\,m^{-3}}]$ using the ideal gas law, where the ambient air temperature and pressure are obtained from the flux tower.

Wind data were collected using two sensor platforms: a fixed flux tower and the drone. Wind data was captured by the 3D sonic anemometer mounted on the tower at a height of $5\,\mathrm{m}$ above the ground. The wind speed and wind direction data were resampled from $10\,\mathrm{Hz}$ to $1\,\mathrm{Hz}$ to match the timestamps of the $CH_4$ sensor. The data from the eddy-covariance sys-

tem was processed at half-hour intervals using EddyPro (Li-Cor) to determine the Obukhov length $L\,[\mathrm{m}]$ and friction velocity $u_*\,[\mathrm{m\,s^{-1}}]$. Using Monin Obukhov Similarity Theory (MOST; see Stull, 1989), we estimate the vertical profile of the mean wind speed $V(z)\,[\mathrm{m\,s^{-1}}]$ and mean eddy diffusivity $K(z)\,[\mathrm{m^2\,s^{-1}}]$, where $z$ is the distance above the ground. Appendix A includes details on the application of MOST.

The drone quantifies wind speed using its onboard sensors to measure resistance during stable hover or flight. This data,

combined with the drone's GPS and inertial measurement unit (IMU), allows for estimations of wind speed and direction by analyzing the accelerations and attitude adjustments needed to counteract the wind's force (Abichandani et al., 2020). Wind data from the drone were obtained from the flight logs using the Flight Reader software.

Given concerns that the additional bulk and weight of the $CH_4$ sensor might affect readings, we performed a correction for wind speed. During the field campaign, the drone hovered for a total of 90 minutes a couple of meters downwind from the

sonic anemometer under various wind speeds and orientations relative to the wind direction. Wind speed data from the drone were corrected through linear regression against the sonic anemometer data (Fig. S1 in Supplementary Material). The wind




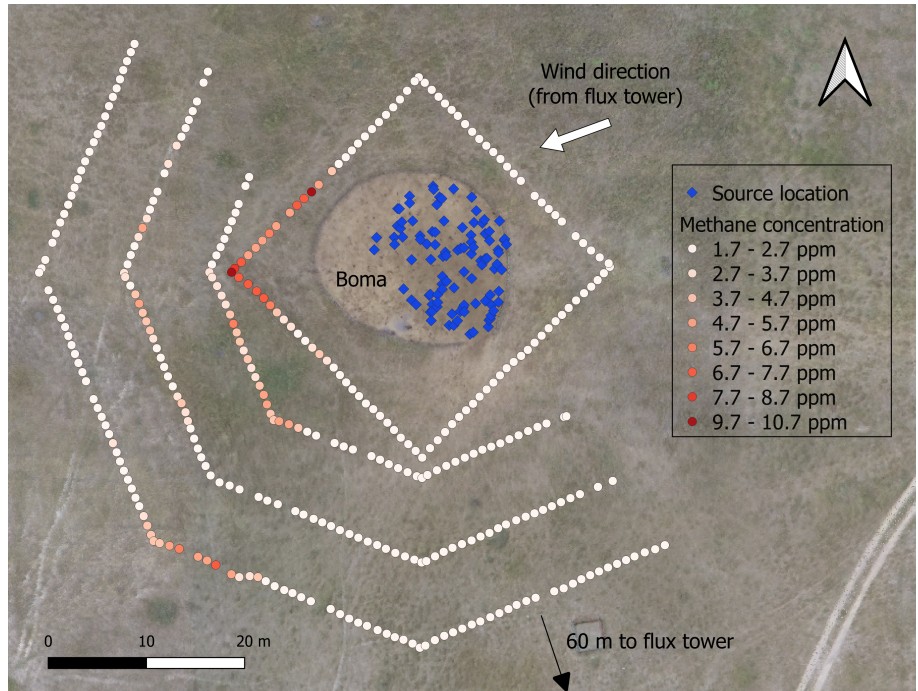

**Figure 2.** Top-down view of the drone flight paths capturing $CH_4$ concentration observations in the afternoon of 6 March 2024. Shown are $CH_4$ concentration measurements obtained at a height of $2.7\,m$ during the mass balance flight around the boma, and at heights of $3.5\,m$, $3.0\,m$, $2.5\,m$ during the half-octagon flight for the Bayesian inference method. The blue markers indicate the source location in the atmospheric dispersion model, representing the heifer herd. The wind direction arrow shows the mean wind direction observed by the flux tower.

direction is reported by the drone in eight compass directions. The wind direction data did not qualitatively match well with the sonic anemometer data and were therefore not used in our study (Fig. S2 in Supplementary Material).

### 2.2.2 Bayesian inference method

The first drone-based method for quantifying $CH_4$ emission rates utilizes an inverse modeling approach, which assimilates atmospheric measurements into an atmospheric transport model to infer emission rates. Two principal approaches are commonly employed in model inversion: (1) Several studies (Andersen et al., 2021; Shah et al., 2019, 2020) minimize a cost function to find the best fit between a Gaussian plume model (Sutton, 1947) and observed $CH_4$ concentrations. (2) In the field of robotics, various studies employ a Bayesian inference approach to model inversion in order to estimate source emission rates and source

locations, among other unknown variables, at local scales (Hutchinson et al., 2017; Francis et al., 2022). Unlike optimization, the Bayesian approach is particularly well suited to solving ill-posed inverse problems involving the assimilation of noisy observations that are ubiquitous in geophysics (Sanz-Alonso, 2023). Beyond robotics, Bayesian frameworks are also utilized





in estimating carbon emissions on regional or global scales from satellite observations (Cusworth et al., 2021; Western et al., 2021) and international ground-based atmospheric observation networks (Evangeliou et al., 2018; Thompson et al., 2022).

We adopt a Bayesian approach, providing a probabilistic interpretation of the model parameters and including uncertainty quantification of the parameter estimates. Previous research has demonstrated the efficacy of Bayesian inference with synthetic drone observations for source localization and estimation (Loisy and Eloy, 2022; van Hove et al., 2024b). However, its applications in real-world environments at a local scale remain relatively limited. Hutchinson et al. (2019) and Park et al. (2021) successfully deployed Bayesian inference methods in outdoor experiments with flat, homogeneous terrain and time-invariant

controlled-release sources, while Hutchinson et al. (2020) explored emissions from a car crash and oil rig at a test site. In real-world conditions, Pirk et al. (2022) assimilated drone observations within a Bayesian framework to infer turbulent fluxes of sensible and latent heat of a wetland and a palsa mire in Norway.

  We use the advection-diffusion model formulated by Vergassola et al. (2007) to simulate $CH_4$ transport under turbulent atmospheric conditions. This model has been shown by Hutchinson et al. (2019) to more accurately represent small-scale

plume behavior compared to the Gaussian plume model. The mean stationary concentration $c\,[\mathrm{g\,m^{-3}}]$ at measurement location $\mathbf{x} = [x, y, z]$ is given by

$$c(\mathbf{x}, \mathbf{x}_s) = \frac{Q/\alpha}{4\pi D|\mathbf{x} - \mathbf{x_s}|} \exp\left(\frac{-(x - x_s)V\sin(\phi)}{2D}\right) \exp\left(\frac{-(y - y_s)V\cos(\phi)}{2D}\right) \exp\left(\frac{-|\mathbf{x} - \mathbf{x_s}|}{\lambda}\right) + c_0 \,, \tag{1}$$

where $\mathbf{x}_s = [x_s, y_s, z_s]$ represents the source location, $Q\,[\mathrm{g\,h^{-1}}]$ denotes the $CH_4$ emission rate, $\alpha = 3600\,\mathrm{s\,h^{-1}}$ is the time conversion factor from hours to seconds, $V\,[\mathrm{m\,s^{-1}}]$ represents the mean wind speed, $\phi\,[°]$ is the mean wind direction, $D\,[\mathrm{m^2\,s^{-1}}]$

denotes the effective diffusivity, $\lambda\,[\mathrm{m}]$ is a characteristic length scale, and $c_0\,[\mathrm{g\,m^{-3}}]$ is the mean stationary background concentration. We make a distinction between the emission rate of the entire herd, denoted by $Q\,[\mathrm{g\,h^{-1}}]$, and the emission rate per individual animal, denoted by $q\,[\mathrm{g\,head^{-1}\,h^{-1}}]$.

  The instantaneous wind fluctuates in amplitude and direction due to effective diffusivity $D$, which is the sum of turbulent diffusivity and the typically much smaller molecular diffusivity. Unlike the Gaussian plume model, which uses dispersion

parameters $\sigma_y$ and $\sigma_z$, typically determined by the stability classification schemes of Pasquill (1961), effective diffusivity $D$ is directly incorporated in the model. Consequently, by making the assumption $D \approx K$, observational estimates of $D$ can be obtained via MOST, as detailed in Appendix A.

  The length scale $\lambda$ in Eq. (1) is defined as

$$\lambda = \sqrt{\frac{D\tau}{1 + \frac{V^2\tau}{4D}}} \,, \tag{2}$$

where $D$ denotes the effective diffusivity, $V$ is the mean wind speed, and $\tau$ is the finite lifetime of $CH_4$ in the atmosphere, approximately 9.1 years (Prather et al., 2012).

  To conserve $CH_4$ mass in Eq. (1), the ground is modeled as a perfect reflector of the plume, as is typically done in Gaussian plume modeling (Hanna et al., 1982). This is achieved by including a mirror image source below the ground surface: $\mathbf{x}_{s,\,\mathrm{mirror}} = -\mathbf{x}_s$. Consequently, the total concentration field becomes the sum of the original and mirrored sources: $c(\mathbf{x}, \mathbf{x}_s) \leftarrow c(\mathbf{x}, \mathbf{x}_s) +$

$c(\mathbf{x}, \mathbf{x}_{s,\,\mathrm{mirror}})$.



In our study, emission rate $Q$, mean wind speed $V$, mean wind direction $\phi$, and effective diffusivity $D$ in Eq. (1) are treated as unknown parameters to be inferred through model inversion. The parameters that are assumed to be known include the source location $\mathbf{x}_s$, drone locations $\mathbf{x}$, and background concentration $c_0$, which have been determined as follows. On the small spatial scale of our study, approximating the herd of animals as a single point source would be an over-simplification. Instead,
we model the herd as a set of $m$ sources, resulting in a total concentration field that is the sum of the individual concentrations from these sources: $c(\mathbf{x}, \mathbf{x}_s) \leftarrow \sum_i^m c(\mathbf{x}, \mathbf{x}_{s,i})$. This source superposition is commonly used in Gaussian plume modeling (e.g. Calder (1977)). Using aerial photographs taken during the drone flights, we randomly select $m = 100$ source locations within the outline of the herd, which together are responsible for emission rate $Q$. For example, the 100 blue markers within the boma shown in Fig. 2 represent the source location of a heifer herd. After inferring emission rate $Q$, we normalize by the
actual number of animals in the herd to obtain the emission rate per individual animal $q$. The source height $z_s$ is estimated by averaging the mouth height of 10 animals from each herd, based on direct measurements. The mean background concentration for each drone flight is empirically determined by calculating the median of the $CH_4$ concentration observations that fall below the threshold of $1.8\,\mathrm{ppm}$.

The drone flew nine legs in a half-octagon pattern downwind of the herd at three different distances: approximately $40\,\mathrm{m}$,
$30\,\mathrm{m}$, and $20\,\mathrm{m}$ from the center of the boma. The corresponding heights for the outer legs were approximately $3.5\,\mathrm{m}$, $5.5\,\mathrm{m}$, and $9.0\,\mathrm{m}$; for the middle legs, $3.0\,\mathrm{m}$, $4.5\,\mathrm{m}$, and $7.0\,\mathrm{m}$; and for the inner legs, $2.5\,\mathrm{m}$, $3.5\,\mathrm{m}$, and $5.0\,\mathrm{m}$ above ground level. To minimize the effects of rotor downwash (visualized with colored smoke in Crazzolara et al. (2019)) and downwind disturbances to the plume (visualized with colored smoke in Hutchinson et al. (2019)), the flights were performed from the outer to the inner legs, starting at the lowermost altitude and ascending to higher altitudes. Figure 2 offers a top-down view of the boma and
illustrates the measured $CH_4$ concentrations along the lowest three legs of the half-octagon flight plan during the drone flight with a herd of heifers on the same day as the satellite overpass.

In our study, we explore the use of three different observing systems for model inversion:

(a) **$CH_4$ concentration data**: We assimilate only instantaneous drone-based $CH_4$ concentration data as observations for $c$.

(b) **$CH_4$ concentration data with drone-derived wind speed**: We assimilate the same concentration data (observation
case (a)) along with mean wind speed observations $V_{\mathrm{obs}}$ derived from drone data. The average wind speed is calculated over the estimated plume depth of $8\,\mathrm{m}$. Specifically, wind speed data from the drone flight is averaged over $1\,\mathrm{m}$ vertical intervals up to $8\,\mathrm{m}$, and the overall average is then obtained over these interval-specific averages.

(c) **$CH_4$ concentration data with flux tower data**: We assimilate $CH_4$ concentration data (observation case (a)) in combination with observations for mean wind speed $V_{\mathrm{obs}}$, mean wind direction $\phi_{\mathrm{obs}}$, and effective diffusivity $D_{\mathrm{obs}}$ derived from
the flux tower data. Hereby, we make the approximation that $D \approx K$, and assume that the vertical and horizontal diffusivity are equal, as is done in Eq. (1). We obtain mean wind speed and diffusivity values by averaging their respective profiles - Eq. (A1) and Eq. (A3) - over the estimated vertical plume extent of $8\,\mathrm{m}$.

We employ a probabilistic approach to model inversion, applying Bayesian inference recursively to mini-batches of observational data to make the problem more computationally tractable (Chopin, 2002). At each new iteration step $n + 1$, the



dynamic prior probability distributions of the unknown parameters $p(\boldsymbol{\theta}|\mathbf{d}_{0:n})$ are updated to the posterior probability distributions $p(\boldsymbol{\theta}|\mathbf{d}_{0:n+1})$ given a new mini-batch of observations $\mathbf{d}_{n+1}$ via Bayes' rule

$$p(\boldsymbol{\theta}|\mathbf{d}_{0:n+1}) = \frac{p(\mathbf{d}_{n+1}|\boldsymbol{\theta})p(\boldsymbol{\theta}|\mathbf{d}_{0:n})}{p(\mathbf{d}_{n+1}|\mathbf{d}_{0:n})},\tag{3}$$

where the conditional model evidence (or marginal likelihood) acts as a normalizing constant

$$p(\mathbf{d}_{n+1}|\mathbf{d}_{0:n}) = \int p(\mathbf{d}_{n+1}|\boldsymbol{\theta})p(\boldsymbol{\theta}|\mathbf{d}_{0:n})d\boldsymbol{\theta}.\tag{4}$$

At each new iteration $n+1$, the dynamic prior distributions $p(\boldsymbol{\theta}|\mathbf{d}_{0:n})$ are simply the corresponding dynamic posteriors from the previous iteration $n$. The use of such sequential Bayesian updating makes the inference problem more computationally tractable and is a key property of the sequential Monte Carlo methods that we employ in practice (Chopin and Papaspiliopoulos, 2020). Note the slight abuse of notation where $\mathbf{d}_0$ is implicitly empty, and thus $p(\boldsymbol{\theta}|\mathbf{d}_0)$ - rather than the usual $p(\boldsymbol{\theta})$ - denotes the initial prior at $n=0$ for notational convenience.

The likelihood term $p(\mathbf{d}_{n+1}|\boldsymbol{\theta})$ in Eq. (3) links the observations to the forward model, effectively serving as a measure of discrepancy between the observed data and the model predictions. The observational model relating observations $\mathbf{d}$ to the forward model prediction is given by

$$\mathbf{d} = \mathcal{F}(\boldsymbol{\theta}) + \boldsymbol{\epsilon},\tag{5}$$

where $\boldsymbol{\epsilon}$ represents the discrepancy (or residual) term, explicitly capturing the various sources of error in the measured data (and

implicitly also errors in the model). For the $CH_4$ concentration observations $c_{obs}$, the forward model $\mathcal{F}$ is defined by Eq. (1). In the case of the wind and diffusivity observations, $\mathcal{F}$ is a more direct noisy mapping; for example, the observed mean wind speed $V_{obs}$ is modeled as $V_{obs} = V + \epsilon_V$.

As Rao (2005) identifies, discrepancies in atmospheric dispersion modeling can arise due to: (a) noise in the sensor measurements, (b) errors in the model input data, (c) the fact that atmospheric dispersion models are imperfect, and (d) inherent

randomness in unresolved turbulent dispersion processes. Given the limited knowledge of these errors, the Gaussian distribution is the most conservative choice for the likelihood function according to the maximum entropy principle (Jaynes, 2003). Thus, we define the likelihood function as a Gaussian of the form $p(\mathbf{d}_{n+1}|\boldsymbol{\theta}) = \mathcal{N}(\mathbf{d}_{n+1}|\mathcal{F}(\boldsymbol{\theta}), \mathbf{R})$ where the mean vector $\mathcal{F}(\boldsymbol{\theta})$ contains the model predictions and $\mathbf{R}$ is a diagonal observation error covariance matrix with observation error variances $\sigma^2$ along the diagonal. These observation error variances correspond to the respective observation error standard deviations empirically

estimated as $\sigma_c = 1\,\mathrm{ppm}$, $\sigma_V = 0.30\,\mathrm{m\,s^{-1}}$, $\sigma_\phi = 10^\circ$, and $\sigma_D = 0.15\,\mathrm{m^2\,s^{-1}}$ for concentration $c$, mean wind speed $V$, wind direction $\phi$, and effective diffusivity $D$, respectively.

We recognize a discrepancy between the timescales of our concentration observations and the statistical assumptions of our dispersion model presented by Eq. (1): while the observations are instantaneous samples of a turbulent boundary layer, our model represents a time-averaged plume. This discrepancy or representation error (Van Leeuwen, 2015) is expected to be the

largest source of uncertainty in our Bayesian inference approach. Additional sources of uncertainty include the assumption of a fixed vertical plume extent, ignoring uncertainties inherent in the (assumed) known variables such as the background





concentration, among other factors. To account for these approximations and minimize the impact of potential errors, we incorporate a high level of uncertainty into the likelihood term by inflating the observation error covariance $\mathbf{R}$ to match the aforementioned observation error standard deviations.

The initial prior distributions $p(\boldsymbol{\theta}|\mathbf{d}_0)$ are chosen to be flat non-informative priors in the form of uniform distributions across defined ranges to match reasonable prior expectations (Banner et al., 2020): $q \sim \mathcal{U}(0.36, 36.00)\,\mathrm{g\,head^{-1}\,h^{-1}}$, $V \sim \mathcal{U}(0, 6)\,\mathrm{m\,s^{-1}}$, $\phi \sim \mathcal{U}(-45, 135)^{\circ}$, corresponding to the wind compass half of the prevailing wind direction, and $D \sim \mathcal{U}(0.3, 3.0)\,\mathrm{m^2\,s^{-1}}$. We implement the Bayesian inference framework in Python (available from van Hove et al. (2024a)) by a Sequential Monte Carlo (SMC) framework (Chopin and Papaspiliopoulos, 2020; Särkkä and Svensson, 2023) that generalizes the classic boot-

strap particle filter (Gordon et al., 1993). This method approximates the probability distributions with a set of weighted ensemble members, referred to as particles. In each iteration, the weights of these particles are updated based explicitly on their likelihood, representing their fit to the observed data, and implicitly on the dynamic prior. To address the particle degeneracy problem, where only a few particles retain significant weights, we apply the resample-move algorithm (Gilks and Berzuini, 2001; Doucet and Johansen, 2009). This algorithm enhances the particle diversity and exploration of the parameter space by

combining resampling with subsequent Markov Chain Monte Carlo (MCMC) moves. Additionally, reflective boundaries are used to respect the predefined ranges of the prior uniform distributions. In our algorithm, we use 25,000 particles, a mini-batch size of 200 observations, and perform five MCMC steps per iteration step.

Due to the inherently stochastic nature of the SMC algorithm, different realizations can yield varying results. This variability arises from randomness in the prior sampling, the generation of proposals in each MCMC step, the selection of the mini-batches

of observations, and the determination of the $m$ source locations representing the herd. As a result, it is common practice to run the SMC algorithm multiple times to (a) assess the variability of its output and (b) obtain more reliable statistical estimates of the inferred parameters (Chopin and Papaspiliopoulos, 2020; Vergé et al., 2015). Thereby, we perform 22 independent realizations of the SMC algorithm in an outer loop to derive more robust estimates of the CH$_4$ emission rates.

### 2.2.3 Mass balance method

The second drone-based method for quantifying CH$_4$ emission rates uses a mass balance approach. Based on the divergence theorem, this technique determines the emission rate from a CH$_4$ source by assessing the net horizontal inflow and outflow of CH$_4$ within an imaginary box enclosing the source. The mass balance approach, or box model, has been widely utilized with drone observations in various studies. For example, Allen et al. (2019) and Gålfalk et al. (2021) estimated emissions from landfills, while Andersen et al. (2021) determined emissions from coal mining ventilation shafts, and Vinković et al. (2022)

investigated emissions from a dairy farm. Additionally, Golston et al. (2018) and Yang et al. (2018) applied the mass balance method with a laser-based CH$_4$ sensor capturing a column-integrated concentration along a vertical path between the drone and the ground to investigate natural gas leaks. On larger scales, the method has been applied using aircraft observations. For example, Cambaliza et al. (2014) assessed emissions of an urban region including multiple sources such as power plants, landfills, and wastewater treatments, while Arndt et al. (2018) quantified emissions of dairy farms encompassing animal housing

and liquid manure storage. On regional or global scales, mass balance analysis is used to estimate emission rates from satellite





observations (e.g. Pandey et al. (2019); Borchardt et al. (2021)). However, Varon et al. (2018) notes that this method is susceptible to large errors. This is due to the inability to accurately parameterize turbulence on the small scale of instantaneous plumes, as well as poor characterization of the vertical wind speed profile between the ground surface and satellite.

The drone collects $CH_4$ point measurements along the vertical planes of an imaginary box encapsulating the source. Data

are then interpolated onto a regular grid to calculate the net emission rate $Q\,[\mathrm{g\,h^{-1}}]$ by integrating the product of $CH_4$ concentration $c\,[\mathrm{g\,m^{-3}}]$ and outward perpendicular wind speed across the vertical sampling planes:

$$Q = \sum_{\text{planes}} \left[ \sum_{i}^{k_i} \sum_{j}^{k_j} c_{i,j}\, v_{\perp\,i,j}\, a_{i,j}\, \alpha \right], \tag{6}$$

where $v_{\perp}\,[\mathrm{m\,s^{-1}}]$ denotes the instantaneous wind speed outward perpendicular to the plane, $a\,[\mathrm{m^2}]$ is the area of a grid cell, $\alpha = 3600\,\mathrm{s\,h^{-1}}$ is the time conversion factor from hours to seconds, $k_i$ is the number of horizontal grid cells of a plane, and $k_j$

is the number of vertical grid cells of a plane.

The vertical sampling planes must be sufficiently high to capture the full extent of the emissions, so that there are only negligible fluxes through the top horizontal plane of the imaginary box. Our imaginary box around the boma measures approximately $26\,\mathrm{m}$ by $26\,\mathrm{m}$ with a height of $10\,\mathrm{m}$. To mitigate downwash effects from the rotors, the drone's flight plan was designed from the ground up. The drone first performed a manual flight at around $1.0\,\mathrm{m}$ above the ground for the lowest leg, followed by

a pre-planned flight mission with ascending legs at approximately $2.0, 2.7, 3.7, 5.2, 7.2$, and $9.2\,\mathrm{m}$. The drone flew at $1\,\mathrm{m\,s^{-1}}$, collecting observations approximately every meter. Figure 2 displays the measured $CH_4$ concentrations at a height of $2.7\,\mathrm{m}$ during the flight with the heifer herd on the same day as the satellite overpass.

To evaluate Eq. (6), we used the horizontal instantaneous corrected wind speed data from the drone and wind direction observations from the sonic anemometer on the flux tower, which we considered more reliable than the drone's wind direction

estimates. Artificial wind speed data points of $0\,\mathrm{m\,s^{-1}}$ were added at ground level to account for the no-slip lower boundary condition. The $CH_4$ concentration and perpendicular wind speeds were interpolated onto north-east (NE), south-east (SE), south-west (SW), and north-west (NW) facing vertical planes. Following the general approach of Gålfalk et al. (2021), we resampled the observations using the following sequence of steps: (a) linear interpolation of the data points onto a regular grid of circa $20\,\mathrm{cm^2}$, (b) averaging onto a coarser grid of circa $1\,\mathrm{m^2}$, and (c) filling any remaining empty grid cells, if any, using

nearest-neighbor values.

The current sampling time of approximately $20\,\mathrm{min}$ is insufficient to capture the mean state of the plume morphology, introducing uncertainty into the emission rate estimate. This primarily stems from temporal variability induced by unresolved atmospheric turbulence affecting wind speed and wind direction ($\sigma_{v,\text{temp}}$ and $\sigma_{\phi,\text{temp}}$). This is further complicated by potential wake effects from the herd that disturb the mean flow field, influencing wind speed observations in the downwind sampling

plane. Additionally, measurement uncertainties in wind speed and wind direction ($\sigma_{v,\text{meas}}$ and $\sigma_{\phi,\text{meas}}$) further contribute to the overall uncertainty in the mass balance approach. Moreover, the wind direction's assessment at the flux tower, rather than at the vertical planes, introduces additional uncertainty. The measurement uncertainty of the $CH_4$ observations is minimal and considered negligible compared to wind-related uncertainties. Similarly, the relative uncertainty of the interpolation process is



considered minor and is excluded from the overall uncertainty estimate. Our approach to uncertainty estimation for the mass
balance method aligns with the practices outlined in Andersen et al. (2021).

We estimate uncertainties due to temporal variation in wind speed and direction ($\sigma_{v,\text{temp}}$ and $\sigma_{\phi,\text{temp}}$) based on their standard
deviation from the mean at the altitude of each leg of the drone flight, following the methodology presented in Cambaliza et al.
(2014). Measurement uncertainties for wind speed and wind direction are estimated at $\sigma_{v,\text{meas}} = 1.7\,\text{m s}^{-1}$ and $\sigma_{\phi,\text{meas}} = 20°$,
respectively. The wind speed uncertainty estimate is derived from the root mean square error between the corrected wind speed
readings from the drone and wind speed records from the sonic anemometer during hovering flights, see Sect. 2.2. The temporal
variation and measurement uncertainty are summed in quadrature: $\sigma_v = \sqrt{\sigma_{v,\text{temp}}^2 + \sigma_{v,\text{meas}}^2}$ and $\sigma_\phi = \sqrt{\sigma_{\phi,\text{temp}}^2 + \sigma_{\phi,\text{meas}}^2}$.
Finally, the total uncertainty estimate for the emission rate is determined through error propagation (Gålfalk et al., 2021;
Andersen et al., 2021; Vinković et al., 2022). A Monte Carlo approach with 500 runs is used for error propagation to incorporate
the various uncertainty sources.

**2.3 IPCC Tier 2 emission values**

In addition to drone-based methods, we estimate CH$_4$ emissions from enteric fermentation of ruminant herds at Kapiti using
the IPCC Tier 2 approach (Paustian et al., 2006). This method is based on the concept of energy balance, where the CH$_4$
emission factor $EF$ [kg head$^{-1}$ d$^{-1}$] of ruminants is calculated by

$$EF = \frac{Y_m\,GE}{E}, \tag{7}$$

where $Y_m$ denotes the fractional methane conversion factor, $GE$ [MJ head$^{-1}$ d$^{-1}$] is the gross energy intake, and $E = 55.65\,\text{MJ kg}^{-1}$
is the energy content of CH$_4$. The daily gross energy intake per animal $GE$ is determined from information on feed quality and
feed intake, live weight of the animals, weight changes, as well as productivity parameters (as specified in Eq. 10.16 in Paustian
et al. (2006)). This data can be obtained at herd level, individual animal level, or a combination of both. Methane conversion
factor $Y_m$ represents the fraction of gross energy intake converted to CH$_4$. The IPCC provides $Y_m$ values for different animal
categories based on review and synthesis of available scientific literature and data (cattle values in Table 10.12, and sheep and
goat values in Table 10.13 of Paustian et al. (2006)). Since specific values are currently unavailable for camels and we do not
have the live weight of the camel herd from our drone flights, we use the IPCC Tier 1 value for camels in our study.

At Kapiti, all livestock herds graze freely during the day, and we assume that their feed intake is entirely from pasture. The
feed quality of the pasture was determined by averaging the nutrient content of 19 samples collected from different locations
across Kapiti on 1 March 2024. These samples were analyzed for dry matter content, nitrogen (converted to crude protein),
carbon, ash, and fibre-fractions (NDF: Neutral Detergent Fibre; ADF: Acid Detergent Fibre; ADL: Acid Detergent Lignin).
This data was used to compute feed digestibility, representing the portion of gross energy intake in the feed not excreted in
feces. Data on average feed intake is difficult to obtain from grazing animals, and was therefore estimated.

The live weight [kg] and time-average daily live weight change [kg d$^{-1}$] of individual animals were determined from direct
measurements taken during the first half of March 2024, and then again at the end of April or May 2024. The average weight and
daily weight change across all animals in a herd were used to compute the $EF$ for the respective animal category. Additional



data used to estimate gross energy intake for each animal herd includes: proportion of pregnant females ($84\%$ of pregnant does herd, $30\%$ of cow herd, and $30\%$ of slick herd); proportion of lactating females ($87\%$ of lactating ewes flock); with average milk production ($1.5\,\mathrm{L\,ewe^{-1}\,d^{-1}}$); number of offspring; and an estimate for the animal's activity, specifically the daily walking distance on the pasture $[\mathrm{km\,d^{-1}}]$. We convert the resulting emission factor $EF$ $[\mathrm{kg\,head^{-1}\,d^{-1}}]$ into emission rate $q$ $[\mathrm{g\,head^{-1}\,h^{-1}}]$ for each animal category to enable method comparison in our study. With the exception of weight and weight changes, all parameters were estimated at the herd level without accounting for associated uncertainties. As we did not perform an explicit uncertainty assessment for the IPCC values, we apply a $\pm 20\%$ uncertainty range for Tier 2 values and a $\pm 30\%$ to $\pm 50\%$ uncertainty range for the Tier 1 value for camels, as reported by the IPCC (Paustian et al., 2006).

## 3 Results and discussion

In this section, we evaluate source location detection through satellite observations and assess each of the drone-based emission estimation methods. We then compare the estimated emission rates from the Bayesian inference method with the results obtained using the mass balance method and IPCC Tier 2 values. A comprehensive overview of the estimated emission rates by the different methods is given in Table B1.

### 3.1 Source detection through satellite observations

Figure 3 shows a true color image of part of Kapiti, captured by the PRISMA satellite on 6 March 2024. Despite the partly cloudy conditions, the irregular cloud cover did not occlude the satellite observations of all three cattle herds within our study area. Maps (c) to (e) show the SR index for a region of 5 by 5 pixels with a spatial resolution of $30\,\mathrm{m}$, across three different sites with (f) five adjacent bomas housing 583 cows, (g) the boma at the drone field site with 206 heifers, and (h) a free-grazing herd of 148 heifers, respectively.

We observe a lower SR index at the herd locations compared to the surrounding background. These low SR anomalies, which indicate relatively low radiance levels in the $CH_4$ absorption feature, may suggest higher atmospheric $CH_4$ concentrations and therefore point to the presence of a $CH_4$ source. The observed lower SR index precisely at the cattle herd locations illustrates the feasibility of using PRISMA satellite imagery to detect the location of potential $CH_4$ sources. In addition to $CH_4$, bare soil exhibits unique features in the SWIR. We detected anomalies with two herds inside a boma against a bare soil background, and a free-grazing herd against a green vegetation background. Although the limited dataset prevents a robust assessment, these preliminary results reinforce our confidence in the effectiveness of our approach for detecting the location of potential $CH_4$ sources. Further dedicated studies are necessary to evaluate the generalizability of these findings.

Super-emitters are characterized by high concentration plumes with large spatial extents, often covering multiple pixels. In contrast, a low SR anomaly of a single pixel can be observed at the location of the cattle herds, as shown in Fig. 3. The absence of a discernible emission plume complicates the estimation of emission rates directly from the satellite data. While emission rates of super-emitters can be quantified from satellite images (Jacob et al., 2022), the current spatial resolution, spectral resolution and precision of satellite observations are insufficient for accurately estimating the emission rates of smaller



## Potential methane anomalies (30 m resolution)

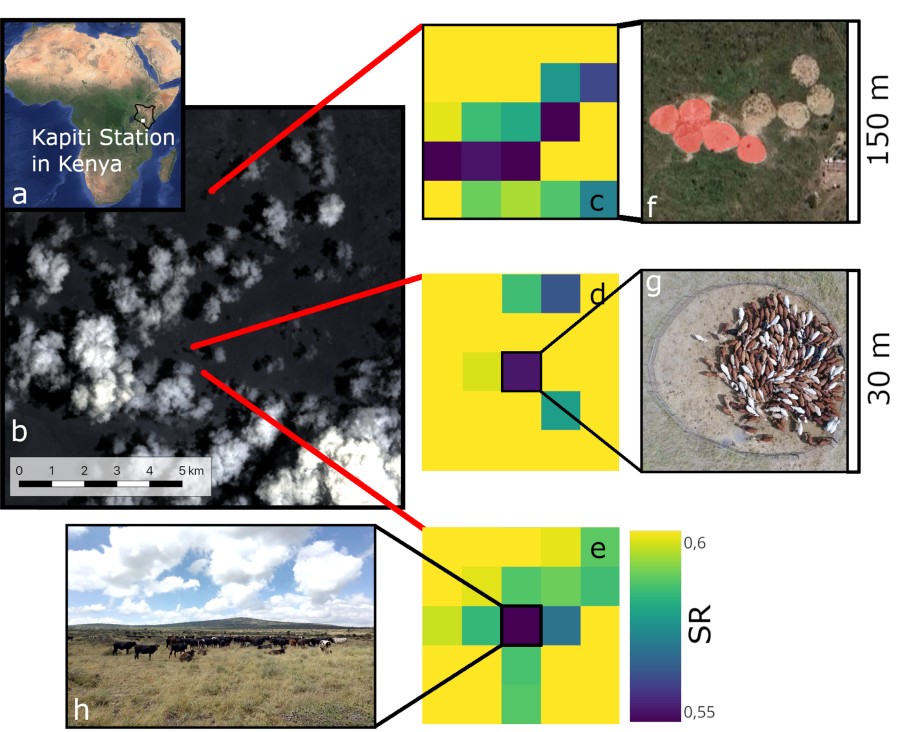

**Figure 3.** (a) Location of the field campaign at the Kapiti Research Station in Kenya. (b) PRISMA true color image of Kapiti from 6 March 2024 08:00 Coordinated Universal Time (UTC) + 3 hours, corresponding to East Africa Time. (c) to (e) The Simple Ratio (SR) radiance index of $2300/2100\,\mathrm{nm}$ with $30\,\mathrm{m}$ resolution for three distinct sites: (f) five adjacent bomas (shaded red) housing 583 cows at the time of the satellite overpass but empty in this picture, (g) a single boma at the drone field site with 206 heifers, and (h) a free-grazing herd of 148 heifers. (a) and (f) include © Google Satellite Imagery (2021). PRISMA product derived from L1 2024-03-06 ©Italian Space Agency (ASI) (2024). All rights reserved.

sources like livestock herds. The quantification threshold of point sources has been found to be $200\,\mathrm{kg\,h^{-1}}$ for GHGSat and 420 $1,400-4,000\,\mathrm{kg\,h^{-1}}$ for Sentinel-2, Landsat-8, PRISMA, and WorldView-3 in controlled release experiments under favorable conditions (Sherwin et al., 2023). In contrast, a herd of 200 cows emits considerably less, roughly $1-4\,\mathrm{kg\,h^{-1}}$ (Broucek, 2014).

Detecting the location of potential $CH_4$ sources could represent a first step in mapping regional $CH_4$ sources. Such mapping can lay the groundwork for applying source term estimation methods, as targeted measurement campaigns can then focus on 425 investigating these identified potential source locations. In our study with ruminant herds, the source locations are already known. However, this approach can be particularly useful for regions like thawing permafrost landscapes, where $CH_4$ source locations are generally unknown. Additional measurements are necessary to verify the location detection of potential $CH_4$



sources based on low SR anomalies and to quantify their emission rates. High-resolution observational platforms, such as drones, are needed to obtain the detailed data required to estimate $CH_4$ sources from livestock herds, such as those at Kapiti.

### 3.2 Source term estimation through drone observations

#### 3.2.1 Bayesian inference method

In the atmospheric dispersion model given by Eq. (1), each of the meteorological parameters influences different aspects of the emission plume. Specifically, the wind direction determines the plume's orientation, while wind speed and diffusivity influence the plume's shape, and the emission rate determines how elevated the plume's concentration level is above the background. We consider these four parameters - wind direction $\phi$, wind speed $V$, diffusivity $D$, and emission rate $q$ - as unknowns to be inferred. We compare Bayesian inference results across three observation cases: (a) using only concentration data, (b) combining concentration data with drone-derived wind speed, and (c) incorporating concentration data with mean wind direction, mean wind speed, and diffusivity obtained from MOST.

Figure 4 presents the Bayesian inference results for the drone flight conducted with the heifer herd on the afternoon of 6 March 2024, on the same day as the satellite overpass. Results for all drone flights are included in Fig. S4 to S24 of the Supplementary Material, and are reported in Table B1. In this section, we analyze the inference results, discussing the overall findings across all drone flights, and examine whether the results presented in Fig. 4 align with these overall findings.

In our study, we frequently observe different patterns when using drone-based methods to estimate $CH_4$ emission rates of sheep and goat herds at Kapiti compared to cattle herds. Except for camels, the herds consist of approximately 100 to 200 animals (see Table B1). Due to the markedly lower emission rate per individual animal $q$ for sheep and goats, these herds have a lower overall emission rate $Q$. Consequently, we refer to the sheep and goat herds as 'weak(er) sources' to denote their relatively lower emission rates in our study.

In most of the drone flights, the inferred mean wind direction aligns with the fixed source location and areas of elevated concentration. Overall, the inferred wind direction is both precise and consistent across all three observation cases for cattle drone flights, occasionally overriding the observed wind direction. In Fig. 4, the mean wind direction estimates across the three observation cases coincide, and the inferred direction of $81°$ corresponds to the angle between the source location and the observation locations with elevation concentration shown in Fig. 2. Specifically, the update in wind direction in observation case (a) indicates that our framework can often infer the wind direction solely from the shape of the concentration plume and the known source location. The posterior mean wind direction becomes more uncertain when dealing with highly variable wind directions without direct observation (observation case (a)), such as during the morning flight with camels (Fig. S14) and the morning flight with pregnant does (Fig. S21).

For weak sources, the Bayesian inference algorithm can misinterpret concentration observations as being upwind of a strong emission source rather than downwind from a weak one if no direct wind direction observation is provided. To address this equifinality issue, we used an informed prior bounded by the half wind-rose: $\mathcal{U}(-45, 135)°$. During the first two days of field work, we used a narrower V-shaped flight path instead of the usual half-octagon. Consequently, for the drone flights on these




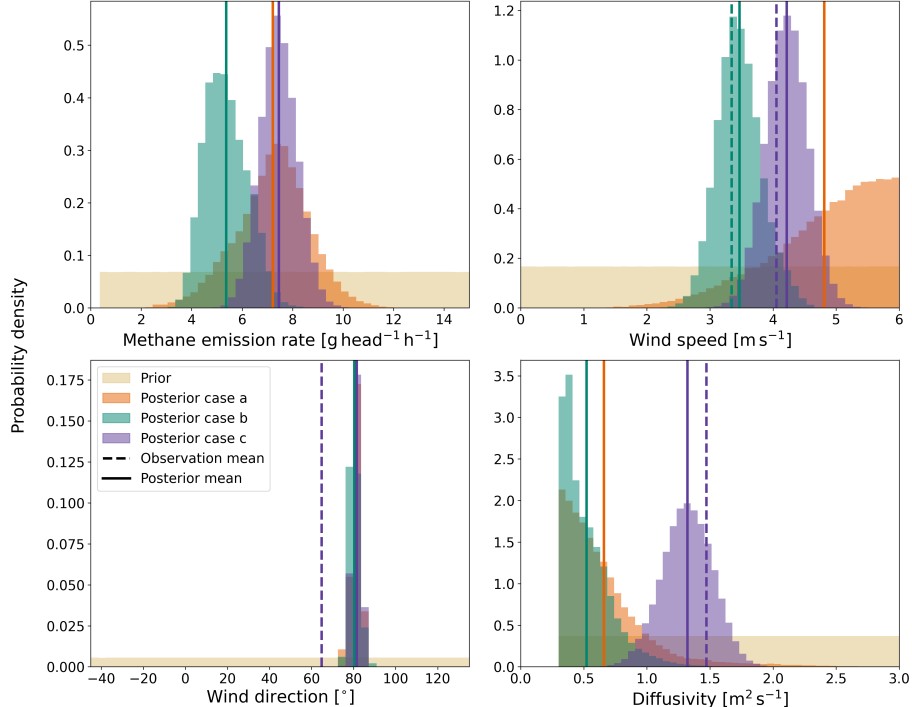

**Figure 4.** Bayesian inference results in the form of posterior distributions obtained using sequential Monte Carlo for the drone flight in the afternoon of 6 March 2024 with 208 heifers. Estimates for three different observation cases: (a) using concentration observations, (b) using concentration observations and mean wind speed data from the drone, (c) using concentration observations and mean wind speed, mean wind direction and diffusivity data derived from MOST.

days, marked with a bullet (•) in Table B1, we adjusted the prior to $\mathcal{U}(30, 135)°$. When dealing with weak sources, an informed prior for wind direction was needed in our study to refine the emission rate distribution for observation cases without direct wind direction observations. Even with an informed prior, the posterior distribution of the wind direction can remain relatively uninformed for weak sources, such as weaner kids (Fig. S23 and S24). We note that concentration observations obtained in all

locations around the source, such as those from the mass balance flight, can help mitigate this ambiguity.

Wind speed and diffusivity influence the shape of the emission plume. Higher wind speeds elongate the plume, while increased diffusivity broadens it. In observation cases (b) and (c), where direct wind speed observations are available, the posteriors generally align with these observed values. Typically, the wind speed derived from MOST (observation case (c)) is higher than the wind speed recorded by the drone (observation case (b)), as demonstrated in Fig. 4. Without direct wind speed

observations (observation case (a)), the Bayesian inference algorithm tends to skew the posterior distribution toward higher wind speeds for most drone flights, as illustrated in Fig. 4. For all drone flights with weak sources, the wind speed posteriors for observation case (a) remain largely uninformed but are slightly skewed toward higher values.



For diffusivity, the posterior distribution in observation case (c) aligns with the direct observation, as shown in Fig. 4. In the absence of observations (observation cases (a) and (b)), the diffusivity posterior remains largely uninformed but slightly

skewed toward higher values in drone flights with weaker sources as well as for the camel herd. For stronger sources, namely the cattle herds, the posterior distribution of diffusivity does often shift toward low values, as shown in Fig. 4.

A relationship was observed between the combination of high posterior wind speeds and high posterior diffusivity, resulting in higher estimated emission rates. Higher wind speed and diffusivity indicate a larger plume, both in length and width, suggesting a larger emission rate provided that the concentration observations are the same. The typically higher wind speeds

derived from MOST, compared to drone wind speeds, combined with higher posterior diffusivity in observation case (c) compared to observation case (b), generally lead to higher emission rates in observation case (c) compared to observation case (b). This is demonstrated in Fig. 4. The observed relation highlights the importance of reliable wind measurements. We consider observation case (a) the least reliable, as the wind speed estimates often skew toward excessively high values. Observation cases (b) and (c) present an interesting topic for future study: Is it more valuable to have an anemometer on the

drone to capture local and temporal wind variations, or to place an anemometer close to the source at a fixed location and use MOST to obtain diffusivity observations? Comparing different observing systems in controlled release experiments with a constant and known emission rate can provide further insights into the optimal experimental set-up for Bayesian inference observing systems.

Across all drone flights, the emission rate estimates for observation case (c) have a smaller relative uncertainty range com-

pared to observation cases (a) and (b). Specifically, the range is approximately $\pm 50\%$ for strong sources and $\pm 12\%$ for weak sources, in contrast to $\pm 65\%$ and $\pm 26\%$ for observation case (a), and $\pm 55\%$ and $\pm 19\%$ for observation case (b), respectively.

We compare our estimates to IPCC Tier 1 values to assess their plausibility. It is important to note that IPCC Tier 1 values are highly uncertain, and we do not use these values as a definitive benchmark but rather as a sanity check. Overall, the Bayesian inference emission rate estimates for both strong and weak sources are of the same order of magnitude as the IPCC Tier 1 val-

ues: $5.3\,\mathrm{g\,head^{-1}\,h^{-1}}$ for dairy cattle in Africa, $5.3\,\mathrm{g\,head^{-1}\,h^{-1}}$ for camels in developing countries, and $0.6\,\mathrm{g\,head^{-1}\,h^{-1}}$ for sheep and goats in developing countries, with a reported uncertainty range of $\pm 30\%$ to $\pm 50\%$ (Paustian et al., 2006). Despite large variations in the posterior estimates for wind direction, wind speed, and diffusivity, the emission rate estimates across all observation cases remain consistent with these reference values. For example, for the drone flight presented in Fig. 4, the emission rate estimates of $7.2\pm1.4\,\mathrm{g\,head^{-1}\,h^{-1}}$, $5.4\pm0.8\,\mathrm{g\,head^{-1}\,h^{-1}}$, and $7.4\pm0.7\,\mathrm{g\,head^{-1}\,h^{-1}}$ for observation cases (a)

to (c) respectively, are considered feasible when compared to the Tier 1 value for dairy cattle in Africa. This underscores the importance of reliable concentration observations, as they alone (observation case (a)) can provide reasonable emission rate estimates.

Further improvement of the Bayesian inference method could involve extending the sampling duration to more accurately capture the time-averaged plume. Prolonging the sampling time at each observation location may require modifications to the

likelihood function. For example, Hutchinson et al. (2019) used a sample duration of $5\,\mathrm{s}$ and applied different likelihoods for concentration observations below and above a plume detection threshold. It is important to consider the trade-off between overall sampling duration and the number of sample locations. Investigating this trade-off, along with the formulation of the





likelihood function, would be a valuable area for future study to improve the Bayesian inference method for estimating $CH_4$ emission rates. Such optimization could maximize the informational value derived from observations collected with a single battery set.

A promising approach to the information maximization strategy involves the use of autonomous drones that can make in-flight decisions about the optimal sampling path based on real-time observations and previously obtained knowledge. Several studies explore this possibility using reinforcement learning (Loisy and Eloy, 2022; van Hove et al., 2024b). However, these studies often rely on synthetic data, while research involving natural $CH_4$ sources under real-world conditions remains limited. Instead of addressing the information maximization strategy solely on the data collection side, exploring the capabilities of Bayesian hierarchical modeling (Berliner, 2003) to enhance the utilization of information in future research is potentially valuable. The hierarchical Bayesian approach allows information to be shared across drone flights, enabling data to be pooled across, for example, the two heifer drone flights in this study, or across all drone flights involving cattle.

### 3.2.2 Mass balance method

Figure 5 presents the mass balance results of the afternoon drone flight with a herd of heifers, on the same day as the satellite overpass. The panels show the interpolated $CH_4$ concentrations (top), perpendicular wind speed (middle), and resulting fluxes (bottom) at the four vertical sampling planes in NE, SE, SW, and NW directions. The sum of the fluxes in the bottom panel equals the final estimated emission rate for the entire herd $Q$. The results of the other drone flights are included in Fig. S25 to S45 in the Supplementary Material, and Table B1.

Figure 2 demonstrates the intermittency of the observed instantaneous plume: the concentrations within the plume do not follow a smooth, continuous gradient but instead exhibit an irregular distribution of disconnected patches of high concentration. This intermittency complicates the mass balance approach, particularly for drone flights where the signal-to-noise ratio is relatively low. Such conditions include: (a) highly variable wind direction or low wind speeds leading to very non-stationary wind conditions (the noise is particularly high), and (b) drone flights with weaker emission sources resulting in low concentration levels (the signal is particularly low) where the variability in the background concentrations and emission plume can considerably affect the accuracy of the emission rate estimate.

The mass balance approach relies on a nonzero horizontal wind to generate a horizontal outflow of $CH_4$ from the imaginary box. Its accuracy improves when the plume morphology remains relatively stable over time. Yang et al. (2018) defines favorable wind conditions as a wind speed greater than $2.3\,\mathrm{m\,s^{-1}}$ and a steady wind direction with a standard deviation below $33.1°$. Measurements collected under wind conditions that do not meet these criteria are marked with a diamond (♦) in Table B1. These sub-optimal conditions lead to less reliable estimates, as for example observed by the negative mission rate of the morning drone flight with camels (Fig. S35). Consequently, these results should be considered unreliable due to unfavorable wind conditions.

Our observations indicate that the estimates for sheep and goats, marked by a triangle (▲) in Table B1, are very variable and inconsistent with the IPCC Tier 1 value of $0.6\,\mathrm{g\,head^{-1}\,h^{-1}}$ (Paustian et al., 2006). Emissions from these smaller animals produce lower $CH_4$ concentration levels, resulting in a lower signal-to-noise ratio when considering the size of the herd. More-



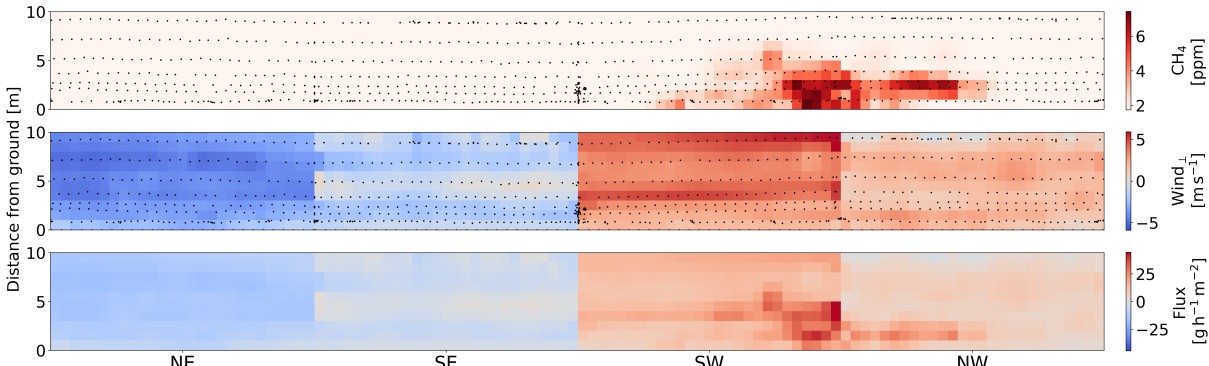

**Figure 5.** Results of the mass balance approach for a drone flight in the afternoon of 6 March 2024 with a herd of 208 heifers. The panels show (top) the interpolated $CH_4$ concentration observations, (middle) the interpolated perpendicular wind speeds, and (bottom) the $CH_4$ fluxes across the vertical sampling planes. Positive (negative) perpendicular wind speeds and fluxes correspond to flow out of (into) the box. The black scatter points indicate the original observation locations.

over, for sheep and goats, the emission source and consequently the plume, are close to the ground in a region characterized by generally lower wind speeds and complex surface effects caused by, for example, variations in elevation and vegetation. This may increase plume variability, leading to less reliable estimates. Due to these factors, we consider the mass balance estimates

for drone flights with sheep and goats unreliable. The mass balance estimates for cattle are more plausible, as they are of the same order of magnitude as the IPCC Tier 1 value of $5.3\,\mathrm{g\,head^{-1}\,h^{-1}}$ for dairy cattle in Africa (Paustian et al., 2006). The negative emission rate for camels in the morning is unrealistic, but the estimate in the afternoon is within a similar range as the IPCC Tier 1 value of $5.3\,\mathrm{g\,head^{-1}\,h^{-1}}$ (Paustian et al., 2006).

     To mitigate the effect of plume variability over time, several studies have conducted repeated drone flights (Gålfalk et al.,

2021; Andersen et al., 2021). In the absence of systematic bias, this approach can yield a more robust approximation of the emission rate by averaging the estimates from multiple drone flights, thereby reducing - though not eliminating - uncertainty due to temporal variability. In a single drone flight, capturing the time-averaged plume can potentially be improved by increasing the number of plume observations relative to background observations. Several studies, particularly larger-scale experiments using airplanes, conduct flights along a single vertical sampling plane downwind from the prevailing wind direction (Allen

et al., 2019; Cambaliza et al., 2014). The term $c_{i,j}$ in Eq. (6) is then replaced by $c_{i,j} - c_0$, where $c_0$ is the estimated background concentration. However, this sampling approach introduces uncertainty due to the estimation of the background concentration, whose variability must be accounted for in the overall uncertainty estimate.

     Although onboard wind measurements with a sonic anemometer are considered ideal (Allen et al., 2019), practical constraints have necessitated the use of nearby weather stations for wind data in several studies (Allen et al., 2019; Nathan et al.,

2015). Morales et al. (2022) demonstrated through controlled release experiments that using wind data from an anemometer close to the source, which captures changing wind conditions during the flight, is more accurate than applying a wind profile through MOST (Eq. (A1)) that only accounts for wind speed variation with altitude. Given that we did not have an anemome-





ter available on the drone, using corrected wind speeds from the flight controller was our best available option. An onboard
anemometer could reduce measurement uncertainty of the instantaneous wind field, but it does not reduce uncertainty due to
temporal uncertainty.

## 3.3 Method comparison

In this section, we evaluate the $CH_4$ emission rate results obtained using the Bayesian inference method by comparing them to
the results derived from other methods and literature values. Figure 6 presents the Bayesian inference results for observation
case (c) alongside estimates from the mass balance approach, a laser spectrometry study previously conducted at Kapiti by Wolz
et al. (2022), and the IPCC emission values.

In Sec. 3.2.1 and 3.2.2, we observed that the Bayesian inference results for all herds are of the same order of magnitude as
the IPCC Tier 1 values. Conversely, while our mass balance results for strong sources (cattle herds) are within the same order
of magnitude as the IPCC Tier 1 values, they are substantially higher than the IPCC Tier 1 values for weak sources (sheep and
goat herds) for a majority of drone flights. This finding is further supported by comparison to the herd-specific IPCC Tier 2
values. We observe that both the Bayesian inference and mass balance results are in the same order of magnitude as the IPCC
Tier 2 values for strong sources of $Q \approx 700 \, \mathrm{g \, h^{-1}}$ to $Q \approx 1{,}500 \, \mathrm{g \, h^{-1}}$ (Table B1). However, a majority of the mass balance
results are substantially higher than the IPCC Tier 2 values for weaker sources of $Q \approx 70 \, \mathrm{g \, h^{-1}}$ to $Q \approx 140 \, \mathrm{g \, h^{-1}}$ (Table B1).
This inconsistency indicates that the source term estimation threshold of the Bayesian inference method is considerably lower
than that of the mass balance method as applied in our study, suggesting that Bayesian inference can be used to estimate weaker
sources, where the mass balance method might fail to reliably estimate sources.

We compare the Bayesian inference estimates to results from previous studies conducted at Kapiti. Our average (pre-
and post-grazing) emission rate estimate for steers was $7.1 \pm 0.7 \, \mathrm{g \, h^{-1}}$, which aligns with a respiration chamber experiment
showing emission rates ranging from $6.7 - 7.7 \, \mathrm{g \, head^{-1} \, h^{-1}}$ depending on diet (Korir et al., 2022b). Our average emission
rate estimate for lactating ewes was $0.8 \pm 0.2 \, \mathrm{g \, head^{-1} \, h^{-1}}$, which overlaps with the emission rate for sheep ranging from
$0.6 - 0.8 \, \mathrm{g \, head^{-1} \, h^{-1}}$ found in a respiration chamber experiment (Mwangi et al., 2023). Our estimate is on the higher end,
which is expected as emissions from lactating animals are generally larger than those from non-lactating animals due to their
increased feed intake to meet the energy demands of milk production (Broucek, 2014). In another respiration chamber experi-
ment, the estimated emission rates from cows ranged from $7.6 - 11.3 \, \mathrm{g \, head^{-1} \, h^{-1}}$ depending on diet (Korir et al., 2022a). In
contrast, our average estimate was notably higher at $15.2 \pm 1.0 \, \mathrm{g \, head^{-1} \, h^{-1}}$.
Wolz et al. (2022) utilized open-path laser spectroscopy with backward Lagrangian stochastic dispersion modeling to esti-
mate nighttime $CH_4$ emissions from a mixed cattle herd across 14 nights in September and October 2019. The resulting mean
emission rates $Q$ were normalized to the equivalent weight of a cow to obtain $q$ for a hypothetical cow herd, rather than nor-
malizing by the number of animals to obtain $q$ for the observed mixed cattle herd. Figure 6 shows the results obtained at 09:00
East Africa Time (EAT), before grazing, and at 00:30 EAT, after grazing. Note that the latter nocturnal measurements were
obtained later than our drone flights, which were conducted in the afternoon. We observe that our Bayesian inference results
for cows are higher, both before and after grazing, compared to estimates from Wolz et al. (2022). This discrepancy could be





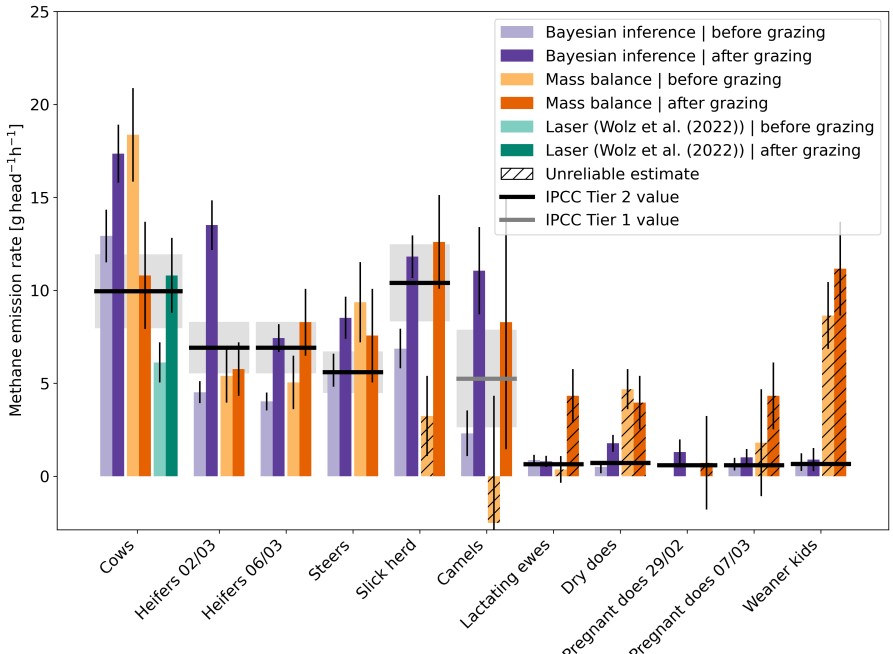

**Figure 6.** Methane emission rate estimates from the Bayesian inference method using concentration observations and mean wind speed, mean wind direction, and diffusivity data derived from Monin Obukhov Similarity Theory (observation case (c)), the mass balance approach, a laser spectrometry study by Wolz et al. (2022), and IPCC emission values. Error bars represent one standard deviation uncertainty. The uncertainty range of IPCC values, depicted by gray shading, is $\pm 20\%$ for Tier 2 values and $\pm 30\%$ to $\pm 50\%$ for Tier 1 values (Paustian et al., 2006), this figure uses $\pm 50\%$ for the IPCC Tier 1 value. Unreliable mass balance estimates due to a low signal-to-noise ratio are indicated by hatched lines.

due to an overestimation by the Bayesian inference method, or the emission rate might have been this high at the times of the drone flights. Wolz et al. (2022) reported a mean emission rate over 14 repeated experiments, whereas the Bayesian inference result is an estimate based on a single drone flight. Additionally, differences in methodology, differences in the timing of the measurements, and the herd weight normalization used by Wolz et al. (2022) could contribute to the variance. Both studies were conducted at the end of a dry season; however, our field campaign was conducted during a normal dry season, whereas the dry season studied by Wolz et al. (2022) was extreme. The severity of this dry season likely affected feed intake and feed quality, potentially reducing $CH_4$ emission rates.

No camel studies have previously been conducted in Kapiti, so we compared our Bayesian inference results for camels to those of a respiration chamber experiment conducted in Australia by Dittmann et al. (2014). This study estimated a $CH_4$ emission rate of $4.0\,\mathrm{g\,head^{-1}\,h^{-1}}$ for Bactrian camels fed exclusively on alfalfa. Our average (pre- and post-grazing) estimate is higher at $6.7 \pm 1.3\,\mathrm{g\,head^{-1}\,h^{-1}}$. Note that the studies involve different camel species and diets. Our average estimate aligns better with the IPCC Tier 1 value of $5.3\,\mathrm{g\,head^{-1}\,h^{-1}}$. We emphasize that the number of experiments conducted with camels



and the extent of current knowledge are minimal, and that further research is required to gain more insight into the emissions
of $CH_4$ from camels. Based on Bayesian inference, we observe a larger increase in estimated $CH_4$ production after grazing in
camels compared to cattle. Replication of this result through repeated experiments would be a promising avenue for further
research.

Feeding is known to increase $CH_4$ production in ruminants (Amon et al., 2001; Hegarty, 2013), and we assess whether this
effect is observable in our results for the other herds as well. The Bayesian inference results show the hypothesized effect of
grazing in seven out of ten cases, with considerably higher emission rate estimates in the afternoon compared to the morning.
In the cases of pregnant does and weaner kids, although there is a general increase in emission estimates before and after
grazing, the considerable overlap in the uncertainty ranges makes the effect less clear. For the final case of lactating ewes, the
difference in emissions before and after grazing is slightly negative; however,the uncertainty ranges overlap, making the effect
inconclusive. In contrast, the mass balance results do not consistently demonstrate an increase in $CH_4$ emissions post-grazing,
with a substantial increase observed in only three out of ten cases - heifers 06/03, slick herd, and lactating ewes. We consider
this to be a promising indicator for the greater reliability and accuracy of our Bayesian inference results compared to our mass
balance results.

With the exception of the cows and lactating ewes drone flights, the Bayesian inference estimates pre-grazing are lower than
the IPCC Tier 2 value, and the post-grazing estimates are higher than the IPCC Tier 2 value, or the uncertainty ranges of the
different methods overlap. One advantage of using the Bayesian inference method over the IPPC Tier 2 approach is the ability
to estimate diurnal variations, which allows us to observe the effects of feeding. Additionally, the uncertainty in the Bayesian
inference results can typically be reduced by assimilating a larger dataset, suggesting that repeated drone flights can yield more
reliable results (Pirk et al., 2022). Deriving IPPC Tier 2 values is time-consuming due to the measurement of live weight and
live weight changes of individual animals, and considerable unquantified uncertainty remains in our ability to estimate the feed
intake of animals in pastoral systems like at Kapiti. Furthermore, the accuracy of the method relies partly on the accuracy of the
methane conversion factor $Y_m$, which is determined based on previous research and may not be representative of the specific
animals being studied.

In comparison to the Bayesian inference method, the mass balance approach is more straightforward to implement. However,
based on this study, Bayesian inference results can be more reliable, as demonstrated by the consistent observation of increased
emissions post-grazing. In both the mass balance method and the Bayesian inference method, we use a sensor - a drone -
to capture snapshots of a non-stationary emission plume. However, the physical models of both methods - Eq. (1) for our
Bayesian inference approach and Eq. (6) for the mass balance method - are based on the assumption of a statistically stationary
plume. A conceptual difference between the two methods lies in how they handle the discrepancy between the turbulent
(instantaneous) observations acquired by the drone and the mean concentration and mean wind field represented in the model.
The Bayesian inference method explicitly accounts for this discrepancy through observation error $\mathbf{R}$, while the mass balance
approach does not explicitly address this inconsistency. Instead, we only account for this violation implicitly by including
a temporal variation term within the uncertainty range of the mass balance estimates. We observed that the mass balance
approach estimates are sensitive to low signal-to-noise levels, making the resulting estimates of weaker sources and under





highly variable wind conditions unreliable. In contrast, the Bayesian inference method proved to be more robust in estimating
weaker sources and under variable wind conditions. This robustness may be attributed to explicitly accounting for discrepancies
between instantaneous observations and the assumed stationarity of the concentration and wind fields in the physical model.

## 4 Conclusions

In our study, we leveraged drone and flux tower observations along with a Bayesian inference approach to quantify $CH_4$ emissions of ruminants in sub-Saharan Africa - an important, yet poorly understood, source in the global $CH_4$ emission inventory.
We showed how this method can be applied to verify and estimate potential $CH_4$ sources identified through radiance anomalies
observed in hyperspectral satellite data.

### 4.1 Bayesian lessons learned

While Bayesian inference is regularly applied on regional and global scales to detect and estimate carbon emissions using
satellite observations and international sensor networks, its application on smaller spatial scales with weaker sources, such as
the livestock herds in our study, is limited. Our work demonstrates the feasibility and effectiveness of using Bayesian inference
methods with drone and flux tower observations for source term estimation on local scales. Here, we share our experiences in
quantifying $CH_4$ emissions using this approach, with the aim of advancing research in local source term estimation through
Bayesian inference.

Rather than treating the herd as a single point source, we modeled it as a set of sources with equal strengths, which we found
to be more accurate. Modeling the herd as a single point source led to higher inferred mean diffusivity in most drone flights
due to the large horizontal spread of instantaneous observed elevated concentrations above background level.

We assumed the set of source locations to be fixed known parameters. When treated as an unknown parameter, the posterior
distribution for source location broadened along the prevailing wind direction and increased the uncertainty in the emission rate
estimates. This can be attributed to equifinality: a stronger source further away can produce a similar concentration observation
as a weaker source nearby. Incorporating concentration data collected around the source enhanced the accuracy of source
location posteriors. We tested this by incorporating the observations of the mass balance flights.

We observed equifinality across multiple parameter combinations, implying that incorporating observations related to unknown parameters is beneficial and may even be necessary to adequately constrain their probability distributions. While wind
direction can be effectively constrained using concentration observations alone, wind speed measurements - such as those from
the drone or flux tower - considerably enhanced the inference process. Further research is recommended to identify the most
reliable observational platform for this purpose. Diffusivity proved difficult to constrain based on instantaneous concentration
observations alone, but we observed that diffusivity observations derived from eddy-covariance data helped to constrain the
diffusivity probability distribution.

We restricted the uniform prior for wind direction to a half wind-rose aligned with the prevailing wind direction. Using the
entire wind-rose introduced ambiguity between upwind observations of strong sources and downwind observations of weak





sources, specifically in drone flights with herds of sheep and goats. Incorporating concentration observations around the source can help mitigate this ambiguity.

Treating the background concentration as a known parameter was necessary in this study because the Bayesian inference algorithm struggled to infer reliable estimates in several drone flights when this parameter was considered unknown, leading
to unreliable emission rate inferences. The algorithm struggled to distinguish between background and elevated concentrations without relying on this assumption. We anticipate that using concentration observations obtained over longer sampling times, with an adjusted likelihood function, could potentially address this issue.

The biggest challenge in setting up the Bayesian framework in our study was the mismatch between instantaneous concentration observations and the time-averaged dispersion model, which complicated the design of the likelihood function. A broad
Gaussian likelihood proved effective. Still, we recommend further investigation into the likelihood design, in combination with extended sampling times to better capture the time-averaged emission plume. However, longer sampling times reduce the number of locations that can be observed on a single drone battery charge. This limitation makes intelligent sampling path design a promising topic for further study.

## 4.2 Insights into $CH_4$ emission detection and estimation from ruminants

By analyzing radiance anomalies in hyperspectral PRISMA satellite data, we detected three cattle herds present at Kapiti Research Station in Kenya. Using the ratio between radiance in the $2300\,nm$ and $2100\,nm$ bands, we showed that it is feasible to locate herds within a boma and free-grazing herds. In particular, this approach allows us to identify the location of potential $CH_4$ sources emitting approximately $1,000\,g\,h^{-1}$ or more. While the detection of the location of potential $CH_4$ sources, such as cattle herds, is feasible using current satellite data, higher-resolution observations are necessary to accurately estimate their
$CH_4$ emission rates.

Using drone observations, we estimated the $CH_4$ emission rates of various ruminant herds - including cattle (cows, heifers, steers, and slick herd), sheep (lactating ewes), goats (dry does, pregnant does, weaner kids), and camels - by applying both Bayesian inference and mass balance approaches. Due to low signal-to-noise levels, the mass balance method did not consistently provide reliable estimates for weaker sources, such as the sheep and goat herds. However, under favorable wind condi-
tions, we estimated cattle herd emissions ranging between $700 - 1,500\,g\,h^{-1}$ using the mass balance method. The Bayesian inference method performed better for weaker sources, estimating sheep and goat herd emissions in the range of $70 - 140\,g\,h^{-1}$, and it was effective for the different cattle and camel herds as well.

We observed the hypothesized increase in $CH_4$ production following feeding in a majority of the Bayesian inference drone flights, whereas this effect was visible in only a minority of the mass balance flights. Additionally, the Bayesian inference
method results appeared less affected by variable wind conditions than the mass balance method. These observations suggest the superior performance of the Bayesian inference method over the mass balance method in this setting.

Overall, the Bayesian inference results aligned with the IPCC Tier 2 emission values. Compared to the IPCC Tier 2 approach, the Bayesian inference method offers the ability to estimate temporal variations in $CH_4$ production. Furthermore, the method can assimilate observations from various measurement platforms and incorporate their uncertainties. The strong performance




of the Bayesian inference approach in this case study, estimating diverse $CH_4$ source strengths under various atmospheric conditions, underscores its robustness and potential as a valuable method for estimating $CH_4$ sources in agricultural areas and other landscapes.

## 4.3  Future applications

We demonstrated the potential of the Bayesian inference method to estimate a range of source strengths. Specifically, we
showed that potential $CH_4$ sources detected by hyperspectral satellites, but too weak to be confidently quantified, can be effectively estimated using drone observations. Future applications of the Bayesian framework for source term estimation could extend to diverse natural and anthropogenic sources, such as $CH_4$ emissions from wetlands, hotspots in thawing permafrost, landfills, and wastewater disposal sites.

In areas where observational data are sparse, the Bayesian inference method can be employed using a gas sensor and an
anemometer mounted on the drone or positioned near the source. This approach could eliminate the need for a nearby flux tower, making the observing framework suitable for remote sites. Conversely, in observation-rich environments, the Bayesian inference method can integrate data from multiple observation platforms. For example, data obtained from laser spectrometry or observations from multiple drones can be assimilated.

Overall, the insights gained in our study demonstrate the potential of Bayesian inference methods, combined with drone
and flux tower observations, for improving our understanding of $CH_4$ emissions at local scales, thereby contributing to the improvement of $CH_4$ inventories and mitigation studies.

*Code and data availability.*  The drone and flux tower data set, along with the Digital Elevation Model of Kapiti, can be accessed from van Hove et al. (2024a). The data processing scripts are available at https://github.com/AlouetteUiO/MIK.

## Appendix A:  MOST

Obukhov length $L$ [m] is positive in stable atmospheric conditions and negative in unstable atmospheric conditions. During all drone flights, the atmosphere was unstable. We use Monin Obukhov Similarity Theory (MOST; see Stull, 1989; Hanna et al., 1982) to estimate the vertical profile of the mean wind speed $V(z)$ [$\mathrm{m\,s^{-1}}$] and mean eddy diffusivity $K(z)$ [$\mathrm{m^2\,s^{-1}}$], where $z$ is the distance above the ground.

Under unstable atmospheric conditions, the mean wind speed profile is estimated by

$$V(z) = \frac{u_*}{\kappa}\left[\ln\left(\frac{z-d}{z_0}\right) - 2\ln\left(\frac{1}{2}\left(1+\frac{1}{\Phi_M}\right)\right) - \ln\left(\frac{1}{2}\left(1+\frac{1}{\Phi_M^2}\right)\right) + 2\arctan\left(\frac{1}{\Phi_M}\right) - \frac{\pi}{2}\right], \quad (A1)$$

where $u_*$ is the friction velocity, $\kappa = 0.4$ is the von Kármán constant, and $z_0$ [m] is the aerodynamic roughness length. We use $z_0 = 0.05\,\mathrm{m}$, which corresponds to terrain with long grass and few trees (Stull, 1989), and a displacement height of $d = 0.10\,\mathrm{m}$.



The dimensionless wind shear $\Phi_M$ is approximated by

$$\Phi_M \left(\frac{z}{L}\right) = \left[1 - 15\frac{z-d}{L}\right]^{-1/4}. \tag{A2}$$

The resulting mean wind speed at the height of the sonic anemometer on the flux tower qualitatively matches the mean wind speeds measured by the same sonic anemometer (Fig. S3 in Supplementary Material).

The eddy diffusivity for effectively passive tracers such as $CH_4$ is generally assumed to be equal to the eddy diffusivity for heat. The mean eddy diffusivity profile is given by

$$K(z) = \kappa u_* \frac{z-d}{\Phi_H}, \tag{A3}$$

where $\Phi_H$ is the dimensionless potential temperature gradient. For unstable atmospheric conditions, it is assumed to be given by

$$\Phi_H \left(\frac{z}{L}\right) = 0.74 \left[1 - 9\frac{z-d}{L}\right]^{-1/2}. \tag{A4}$$

**Appendix B:  Results table**

*Author contributions.*  Conceptualization: AvH, NP, KA. Methodology: AvH, KA, NP. Software: AvH. Formal analysis: AvH (drone-based
methods), VL (IPCC Tier 2 values), RC (satellite data). Investigation: AvH and JH (drone flights), VO (eddy-covariance data), RC and FF (satellite data). Resources: CA (host at Kapiti and coordination field campaign). Writing - Original Draft: AvH. Writing - Review & Editing: KA, NP, VL, CA, RC, VO, FF, AvH. Visualization: AvH, RC (Fig. 3). Funding acquisition: VL, NP.

*Competing interests.*  The authors declare that they have no conflict of interest.

*Acknowledgements.*  We would like to thank Ilona Gluecks from ILRI for providing information about the different ruminant herds at Kapiti,
and Endale Balcha Gurmu from ILRI for assisting with the IPCC Tier 2 workflow. Furthermore, we would like to thank Nehemiah Kimengich, Nelson Kipchirchir and Elly Kibira at Kapiti for their assistance in the field. Moreover, we would like to thank Luc Girod from UiO for his help with drone imagery processing. This study contains modified data of the PRISMA satellite of ESA - Agenzi Spaziale Italiana (ASI) [Year 2024]. We would like to thank Monica Pepe from CNR and Patrizia Sacco from ASI for the data procurement. This work was supported by the Research Council of Norway (projects #333232 (CircAgric-GHG) and #301552 (Spot-On)) and the European Research Council (project
#101116083 (ACTIVATE)). This work is a contribution to the strategic research initiative LATICE (#UiO/GEO103920), the Center for Biogeochemistry in the Anthropocene, as well as the Center for Computational and Data Science at the University of Oslo. This study was supported by the CGIAR Initiatives Livestock and Climate and Mitigate+: Low-Emission Food Systems, which are supported by contributors



to the CGIAR Trust Fund. This work was supported by the European Union through the EU-DeSIRA ESSA project (Earth observation and environmental sensing for climate-smart sustainable agropastoralism ecosystem transformation in East Africa). The content of this article is

the sole responsibility of the authors and does not necessarily reflect the views of the European Union.



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





**Table B1.** Overview of methane emission rate estimates obtained by the mass balance method, the Bayesian inference method, and the IPCC Tier 2 workflow with $\pm20\%$ uncertainty range (Paustian et al., 2006). The cases are defined as: (a) using concentration observations, (b) using concentration observations and mean wind speed data from the drone, (c) using concentration observations and mean wind speed, mean wind direction and diffusivity data derived from Monin Obukhov Similarity Theory. Emission rate estimates are given in $q\,[\mathrm{g\,hd^{-1}\,h^{-1}}]$, where hd is short for head, except for the last column which presents estimates in $Q\,[\mathrm{g\,h^{-1}}]$, i.e. for the entire herd. One standard deviation uncertainty is reported. Dates of drone flights are denoted by [DD/MM]; AM and PM indicate drone flights conducted in the morning before feeding and in the afternoon after feeding, respectively. Wind speed ($V_{EC}\,[\mathrm{m\,s^{-1}}]$) and wind direction ($\phi_{EC}\,[°]$) data are from the flux tower during the mass balance flight. Further notation: (◆) Unreliable estimates due to a low signal-to-noise ratio caused by a weak emission source; (▲) Unreliable drone flight results due to unfavorable wind conditions (following Yang et al. (2018)); (★) IPCC Tier 1 value with $\pm50\%$ uncertainty range: $\mathcal{U}(30, 135)$; (★) IPCC Tier 1 value with $\pm50\%$ uncertainty range (Paustian et al., 2006).

| Date | Herd | | Wind | | Mass balance | Bayesian inference | | | IPCC Tier 2 value | |
|---|---|---|---|---|---|---|---|---|---|---|
| | Animals | Count | $V_{EC}$ | $\phi_{EC}$ | $q$ | $q$, case a | $q$, case b | $q$, case c | $q$ | $Q$ |
| [DD/MM] | [-] | [-] | [$\mathrm{m\,s^{-1}}$] | [°] | [$\mathrm{g\,hd^{-1}\,h^{-1}}$] | [$\mathrm{g\,hd^{-1}\,h^{-1}}$] | [$\mathrm{g\,hd^{-1}\,h^{-1}}$] | [$\mathrm{g\,hd^{-1}\,h^{-1}}$] | [$\mathrm{g\,hd^{-1}\,h^{-1}}$] | [$\mathrm{g\,h^{-1}}$] |
| 02/03 AM | cows | 101 | 3.5±0.9 | 27±24 | 18.3±2.5 | 18.7±4.7 | 11.9±1.7 | 13.0±1.4 | 10.0±2.0 | 1,005±201 |
| 02/03 PM | cows | 101 | 5.2±1.5 | 56±16 | 10.6±3.1 | 15.1±2.8 | 13.0±1.5 | 17.4±1.5 | 10.0±2.0 | 1,005±201 |
| 02/03 AM | heifers | 208 | 3.7±0.8 | 61±21 | 5.2±1.3 | 6.5±1.7 | 4.4±0.8 | 4.5±0.6 | 6.9±1.4 | 1,438±288 |
| 02/03 PM | heifers | 208 | 5.7±1.3 | 66±11 | 5.9±1.4 | 13.3±3.0 | 12.0±2.0 | 13.5±1.3 | 6.9±1.4 | 1,438±288 |
| 06/03 AM | heifers | 206 | 3.9±0.9 | 48±13 | 5.1±1.3 | 4.6±1.3 | 3.1±0.9 | 4.0±0.5 | 6.9±1.4 | 1,425±285 |
| 06/03 PM | heifers | 206 | 5.8±1.5 | 70±14 | 8.2±1.7 | 7.2±1.4 | 5.4±0.8 | 7.4±0.7 | 6.9±1.4 | 1,425±285 |
| 05/03 AM | steers | 127 | 4.9±1.1 | 61±12 | 9.5±2.1 | 6.1±1.6 | 4.0±0.8 | 5.7±0.9 | 5.6±1.1 | 710±142 |
| 05/03 PM | steers | 127 | 6.7±1.6 | 53±13 | 7.4±2.5 | 8.1±2.3 | 7.5±1.3 | 8.5±1.1 | 5.6±1.1 | 710±142 |
| 03/03 AM | slick herd | 148 | 2.0±0.9 | 354±59 | 3.3±2.2◆ | 11.4±3.8 | 6.9±2.4 | 7.0±1.1 | 10.4±2.1 | 1,539±308 |
| 03/03 PM | slick herd | 148 | 4.9±1.4 | 55±23 | 12.5±2.1 | 12.5±3.6 | 7.5±1.2 | 11.9±1.1 | 10.4±2.1 | 1,539±308 |
| 04/03 AM | camels | 42 | 1.7±0.7 | 289±47 | −2.6±7.4◆ | 3.4±2.3 | 2.8±1.7 | 2.3±1.2 | 5.3±2.6★ | 231±116★ |
| 04/03 PM | camels | 42 | 4.3±1.4 | 58±28 | 8.4±6.9 | 12.9±4.5 | 8.8±2.8 | 11.0±2.3 | 5.3±2.6★ | 231±116★ |
| 07/03 AM | lactating ewes | 173 | 3.1±1.3 | 83±26 | 0.2±0.7▲ | 0.9±0.4 | 0.8±0.3 | 0.9±0.3 | 0.6±0.1 | 112±22 |
| 07/03 PM | lactating ewes | 173 | 4.3±1.4 | 90±26 | 4.2±1.4▲ | 0.8±0.4 | 0.7±0.3 | 0.8±0.3 | 0.6±0.1 | 112±22 |
| 01/03 AM | dry does | 197 | 2.4±0.7 | 60±35 | 4.8±1.3◆,▲ | 0.8±0.5● | 0.7±0.4● | 0.5±0.3 | 0.7±0.1 | 141±28 |
| 01/03 PM | dry does | 197 | 4.6±1.3 | 46±21 | 3.9±1.5▲ | 1.7±0.7● | 1.5±0.6● | 1.8±0.5 | 0.7±0.1 | 141±28 |
| 29/02 PM | pregnant does | 124 | 5.6±1.3 | 51±17 | 0.8±2.7▲ | 1.3±0.8● | 1.1±0.6● | 1.3±0.7 | 0.6±0.1 | 74±15 |
| 07/03 AM | pregnant does | 124 | 1.4±0.6 | 12±67 | 1.7±1.5◆,▲ | 0.8±0.6 | 0.6±0.4 | 0.7±0.3 | 0.6±0.1 | 74±15 |
| 07/03 PM | pregnant does | 124 | 4.2±1.3 | 62±28 | 4.2±1.9▲ | 1.2±1.2 | 1.2±1.0 | 1.0±0.5 | 0.6±0.1 | 74±15 |
| 01/03 AM | weaner kids | 118 | 3.6±1.1 | 58±15 | 5.0±1.9▲ | 0.8±0.6● | 0.6±0.4● | 0.7±0.5 | 0.7±0.1 | 79±16 |
| 01/03 PM | weaner kids | 118 | 6.2±1.4 | 49±12 | 11.1±2.3▲ | 0.8±0.6● | 0.8±0.6● | 0.9±0.6 | 0.7±0.1 | 79±16 |