# Peer review of "Inferring methane emissions from African livestock by fusing drone, tower, and satellite data"

_EGUsphere, 2024_

## Author Comment (AC1)

**Reviewer # 1**

We sincerely appreciate the reviewer for their careful reading of our manuscript and for providing insightful and constructive feedback. We would like to thank the reviewer for their thoughtful comments and valuable suggestions.

In the following responses, we address each comment systematically. We also include a revised version of the manuscript that highlights all changes.

**Comment # 1.1**

*The authors present a comprehensive study on the estimation of methane emission rates from livestock in sub-Saharan Africa using a combination of drone, flux tower, and satellite data. They developed a Bayesian inference method to assimilate these observations into an atmospheric dispersion model and compared the methane emission rates of various livestock species, including cattle, goats, sheep, and camels, from the Bayesian inference method with estimates from a mass balance approach and IPCC Tier 1 and 2 approaches. The Bayesian inference method was found to be more robust at quantifying emissions from weaker sources than a mass balance approach. The results indicate that the Bayesian inference method is effective in quantifying methane emissions from different livestock species, with promising implications for improving emission inventories, especially for less studied livestock species such as camels.*

**Reply:**

We thank the reviewer for their positive feedback on our manuscript.

**Comment # 1.2**

*In general, the paper is well prepared and well-reasoned. However, there are areas where the manuscript could be improved to enhance clarity and impact. The authors have studied the topic from many angles, and thus the manuscript would benefit from better structure and reorganization of some topics.*

**Reply:**

We appreciate the reviewer's constructive feedback. We have carefully considered the suggestions regarding its structure and organization. Specific responses addressing these points can be found in our replies to Comments #1.3, #1.4, and #1.5

COMMENT # 1.3

*In particular, I wondered about the value of using the PRISMA observations. It seemed to me that the development of the Bayesian inference approach was the main thing in this paper. The use of satellite observations felt a bit like an afterthought, especially since the authors could not use it as a validation of the Bayesian inference approach. Thus, I would suggest that the authors change the order of the paper so, that the sections discussing PRISMA observations would be the last, and that they would be introduced as a demonstration how it could potential be used to verify or upscale the Bayesian inference approach, similar how you state in P3, L77-80: "We apply two distinct methods: a traditional mass balance method and an innovative Bayesian inference approach that uses a sequential Monte Carlo method to invert an atmospheric diffusion model. To complement this analysis, we assess the capability of hyperspectral satellite data to pinpoint the location of CH4 sources, specifically ruminant herds, by identifying spectral anomalies at the landscape level."*

**Reply:**

We thank the reviewer for their thoughtful suggestion regarding the organization of the manuscript. We agree that moving the section on satellite observations after the drone study improves clarity and coherence. We have implemented this reorganization and emphasized how the satellite analysis can complement the Bayesian inference approach using drones. The specific changes throughout the manuscript can be found in the revised version of the manuscript provided below this list of comment responses.

COMMENT # 1.4

*I was also puzzled by the "4.1 Bayesian lessons learned" section. It was under the conclusion but introduced topics that were not properly discussed in the paper. It seems that the authors did a lot of testing with the models before settling on the setup explained in the manuscript. Such testing is desirable, and I appreciate that they report what worked and what did not. However, I think it should be discussed before the conclusions, or at least there could be a mention somewhere at the beginning that the lessons learned will be discussed at the end of the manuscript, for example similar to Saunois et al, 2024 (https://doi.org/10.5194/essd-2024-115). I would keep the conclusion section itself rather short and focused.*

**Reply:**

We thank the reviewer for their insight regarding the placement of the '4.1 Bayesian lessons learned' section. We acknowledge that this section was not optimally placed within the manuscript. As suggested by the reviewer, we have moved the section

to the end of Results & Discussion, prior to the Conclusions. We have included a reference to this section in Materials & Methods to provide better context. The reorganized and revised manuscript can be found below this list of comment responses.

**Changes:**

**2.1.2 Bayesian inference method**

…

The methodological challenges encountered during the implementation of the Bayesian inference method are discussed in Section 3.4.

COMMENT # 1.5

*I would also consider separating the Results and Discussion sections into two different ones.*

**Reply:**

We appreciate the reviewers suggestion to separate the sections. However, we believe that keeping them combined improves the coherence of the analysis, as it allows us to discuss the findings presented in the figures and table immediately.

COMMENT # 1.6

*Specific comments*

*P1, L17-18: "improvement of climate models": I did not find a proper reasoning for how the methods developed in the paper would improve climate models. From my point of view, the major improvements would be within inventories and verifying other methods which estimate methane emissions. Please consider removing this or add reasoning for this.*

**Reply:**

We appreciate the reviewer's comment on our manuscript. We intended to convey that these improvements could ultimately aid in calibrating climate models by improving the accuracy of methane source representation within the model. However, to maintain focus in the abstract, we have removed this part of the sentence and propose the following revision.

**Changes:**

**Abstract**

. . .

The Bayesian observing system could thereby contribute to the  improvement of  emission inventories and verification of other emission estimation methods.

COMMENT # 1.7

*P2, L27: "a mass balance method with drone observations": You mention later that this is an established method to estimate emissions using drone observation. You could mention it already here to highlight that the Bayesian approach is the novel aspect here.*

**Reply:**

We thank the reviewer for their valuable suggestion. We propose to clarify this distinction throughout the manuscript as follows.

**Changes:**

**1 Introduction**

. . .

We assess the efficacy and robustness of this novel method by comparing our results with those obtained from a conventional mass balance method and IPCC emission values.

. . .

To compare the results obtained from the novel Bayesian inference method with estimates from a more traditional mass balance method . . .

. . .

**2 Materials and methods**

. . .

source term estimation through drone observations using an innovative particle-based Bayesian inference method . . . source term estimation through drone observations using a traditional mass balance approach . . .

. . .

**3 Results and discussion**

In this section, we compare the estimated emission rates derived from the novel Bayesian inference method with those obtained using a conventional mass balance method and IPCC Tier 2 values.

COMMENT # 1.8

*P2, L29: "Global mean atmospheric CH4 concentrations surpassed 1.90 ppm in 2022": You could update this value, especially since the study takes place in 2024.*

**Reply:**

We thank the reviewer for their comment. We have updated the value to 1.93 ppm, reflecting the data for 2024. The percentage increase remains at 16% since 1985.

**Changes:**

**1 Introduction**
Global mean atmospheric $CH_4$ concentrations  reached 1.93 ppm in 2024, marking a 16% increase since 1985 . . .

COMMENT # 1.9

*P3, L65: "On much larger spatial scales": Much larger scales than herds and farms? Please, specify.*

**Reply:**

We thank the reviewer to bringing this to our attention. To clarify, methods for detecting and quantifying methane sources from satellite data generally consider emission plumes extending over multiple pixels. We have added a typical spatial scale range for better understanding in the manuscript.

**Changes:**

**1 Introduction**
. . .
On  larger spatial scales, typically spanning several kilometers, satellite observations are used to detect and quantify $CH_4$ emissions from super-emitters, such as leaks from oil and gas production and large landfills . . .

COMMENT # 1.10

*P4, section "2.1 Satellite observations for source detection": I would wish to have a better motivation why you are using observations from PRISMA and not from other satellites. Has it something to do with the spatial resolution and revisit time? Did you check if other satellites had observations over the study areas during the study period? Please, add at least a short description of the PRISMA satellite (spatial resolution and revisit time etc.).*

**Reply:**

We chose to use PRISMA data because our project team had access to this satellite's observations, and the satellite overpass coincided with our drone campaign. This allowed us to accurately note the positions of the herds at the time of the satellite overpass. We have included the following additional details about the PRISMA satellite in the manuscript.

**Changes:**

**2.3 Satellite observations for source detection**

...

On 6 March 2024, the PRecursore IperSpettrale della Missione Applicativa (PRISMA) satellite (Loizzo et al., 2018) was commissioned to capture a hyperspectral image of Kapiti specifically for this research, while three cattle herds were present at different  locations.

The PRISMA satellite has two hyperspectral sensors that  measure solar radiation reflected by the Earth over 240 spectral bands, ranging from 400 nm to 2500 nm. Its spatial coverage is $30 \times 30$ km$^2$, with a resolution of 30 m.

COMMENT # 1.11

*P4, L120: What is a "data cube"?*

**Reply:**

We have added an explanation of what a 'data cube' is to the manuscript for clarification.

**Changes:**

**2.3 Satellite observations for source detection**

...

We processed the hyperspectral data of the PRISMA satellite in the infrared region to detect spatial variations in the CH$_4$ absorption feature. Starting from Level-1 top-of-atmosphere radiance narrowbands (Giardino et al., 2020),  we integrated the infrared information into single data cubes - composite images representing the same pixel across adjacent spectral bands - using the PRISMAread package in the R environment (Busetto and Ranghetti, 2020). This produced hyperspectral data cubes consisting of 173 spectral bands, spanning infrared wavelengths from 920 nm to 2505 nm.

COMMENT # 1.12

*P5, L139: You could mention already here why you have "one flight for each emission esti-mation method" and why the same flights cannot be used for both methods, i.e., they need different flight paths.*

**Reply:**

We thank the reviewer for their question as it raises a topic of interest to us. We believe that flight path design is a valuable area for future research, and mention that "...makes intelligent sampling path design a promising topic for further study" in the Bayesian lessons learned section. For this study, we based our flight plans on prior knowledge. We have included information about the flight paths at the suggested location in the manuscript.

**Changes:**

**2.1 Drone-based source term estimation**

…

Typically, four flights were conducted for each ruminant herd. In the morning, be-fore grazing, two flights were performed: one flight for each emission estimation method  . The Bayesian inference flight focused on monitoring the downwind emission plume at various distances and altitudes from the source. In contrast, the mass balance flight, following Gålfalk et al. (2021), involved flying a virtual box around the source, capturing the differences in concentrations upwind and downwind from the source.

COMMENT # 1.13

*P7, L184: What is "model inversion"?*

**Reply:**

To clarify this concept, we have added an explanation in the manuscript.

**Changes:**

**2.1.2 Bayesian inference method**

…

The first drone-based method for quantifying $CH_4$ emission rates utilizes an in-verse modeling approach . Model inversion involves estimating unknown input parameters of a theoretical model by utilizing observed data related to output parameters. We assimilate atmospheric measurements into an atmospheric

transport model to infer emission rates.

COMMENT # 1.14

*P7, L185: "Unlike optimization": the word "optimization" is also used with Bayesian statistics and can include minimizing a cost function (see for example the references at the end of the chapter).*

**Reply:**

We thank the reviewer for pointing out this oversight in our wording. We propose to revise the manuscript as follows.

**Changes:**

**2.1.2 Bayesian inference method**
...
Two principal approaches are commonly employed in model inversion: (1) Several studies (Andersen et al., 2021; Shah et al., 2019, 2020) minimize a cost function to find the best fit between a Gaussian plume model (Sutton, 1947) and observed $CH_4$ concentrations, using a frequentist framework. (2) In the field of robotics, various studies employ Bayesian inference for model inversion to estimate source emission rates and source locations, among other unknown variables, at local scales (Hutchinson et al., 2017; Francis et al., 2022). The Bayesian approach is particularly well suited to solving ill-posed inverse problems involving the assimilation of noisy observations that are ubiquitous in geophysics (Sanz-Alonso, 2023).

COMMENT # 1.15

*P9, L233: "the threshold of 1.8 ppm": This sounds a bit low compared to the global average (above 1.91 ppm in 2024, `https://gml.noaa.gov/ccgg/trends_ch4/`). How did you settle on this value? You could also discuss more how the chosen background value affected the emission rate estimates.*

**Reply:**

The measured background concentrations in Kenya were lower than our background measurements in Norway before and after the field campaign, where we measured approximately 2.0 ppm, thereby ruling out severe drift in the instrument. Contact with the manufacturer confirmed that the discrepancy in background concentration cannot have been caused by atmospheric pressure differences. Furthermore, the manufacturer cannot detect any abnormalities in our data files or instrument settings.

This leaves us to believe that the measured low background concentration was natural.

We note that potential bias in the sensor readings does not impact the emission rate estimates in either drone method, as both methods rely on the elevated concentrations above the background level.

To provide additional clarity on how we settled on the chosen threshold value and its implications, we propose the following revisions in the manuscript.

**Changes:**

**2.1.2 Bayesian inference method**

...

The peaks of the concentration histogram for each drone flight were below 1.8 ppm, and the mean background concentration $c_0$ was determined as the median of $CH_4$ observations below this threshold. Although this value is lower than the global average of 1.93 ppm, measurements in Norway before and after the field campaign were close to the global average, and data inspection by the manufacturer revealed no sensor abnormalities, suggesting that this low value was natural. Addressing any remaining concerns, we note that both drone methods rely on concentration elevations above the background level, ensuring that any potential systematic bias does not affect the final estimates.

COMMENT # 1.16

*P12, L333: "with a height of 10 m": How did you settle on this height?*

**Reply:**

We selected a height of 10 m because we anticipated that this elevation would position the drone above the plume, allowing us to neglect any fluxes through the top horizontal plane (i.e., lid or roof) of the imaginary box. We obtained observations of the top horizontal plane to ensure that any fluxes there were negligible. If necessary, we could have incorporated the resulting fluxes from those measurements into the mass balance method. However, during data analysis, we confirmed that the drone remained above the plume for all flights, making it unnecessary to include the top horizontal plane in the mass balance method.

COMMENT # 1.17

> *P14, section "3.1 Source detection through satellite observations" As I mentioned before, I think this could be the last section. In addition, to make it more relevant, you could think of adding a simple correlation calculation, i.e. calculate how much each PRISMA pixel should have methane emissions based on the paper (emission rate x number of animals) and then see if there is any correlation between the SR anomalies. At least you could speculate more on how these satellite observations could be used to upscale the results from this paper.*

**Reply:**

We would like to thank the reviewer for their valuable suggestion. We agree that this section is more appropriate after the drone study, and we have revised the manuscript accordingly.

The primary goal of the PRISMA analysis is to explore whether the last generation hyperspectral imagery can detect spectral features associated with CH4 emissions, thereby aiding source term estimation with drones in regions where source locations are unknown. However, since this initial exploration does not provide sufficient evidence to establish a direct correlation between the spectral anomalies and CH4 emissions - given that these anomalies may be influenced by other landscape features - we prefer to refrain from conducting a correlation analysis at this stage. We believe this topic would be better suited for future research with dedicated datasets from multiple sites.

To clarify this point in the manuscript, we propose the changes listed below. In the revision we emphasize the need for further research and highlight how the satellite observations can be used. We believe that these adjustments align well with the new position of the satellite analysis in the manuscript.

**Changes:**

**Abstract**

…

…

Finally, we show that radiance anomalies identified in hyperspectral satellite data can be used to inform targeted drone missions, as these anomalies may serve as indicators of potential methane sources.

…

 2.3 **Satellite observations for source detection**

…

[revised manuscript text omitted]

COMMENT # 1.18

> *P16, L432-436: "Specifically, the wind direction determines the plume's orientation, while wind speed and diffusivity influence the plume's shape, and the emission rate determines how elevated the plume's concentration level is above the background. We consider these four parameters - wind direction $\phi$, wind speed V, diffusivity D, and emission rate q − as unknowns to be inferred." This could be mentioned already in the methods.*

**Reply:**

We thank the reviewer for their feedback. We have moved this information to the Methods & Materials section.

**Changes:**

**2.1.2 Bayesian inference method**

...

Each of the meteorological parameters in Eq. (1) influences different aspects of the emission plume. Specifically, the wind direction determines the plume's orientation, while wind speed and diffusivity influence the plume's shape, and the emission rate determines how elevated the plume's concentration level is above the background. The instantaneous wind fluctuates in amplitude and direction due to turbulent forces, which are influenced by the effective diffusivity D. The effective diffusivity is the sum of turbulent diffusivity and the typically much smaller molecular diffusivity.

COMMENT # 1.19

> *P16, L443: "we frequently observe different patterns": Could you specify, which patterns?*

**Reply:**

By 'patterns', we were referring to the distinct trends that we observed in the results and which are discussed in this section. We propose the following clarification.

**Changes:**

**3.1.1 Bayesian inference method**

...

We frequently observe  distinct trends when using drone-based methods to estimate $CH_4$ emission rates of sheep and goat herds [...] to denote their relatively lower emission rates in our study. These observed distinct trends will be discussed in greater detail in this section.

COMMENT # 1.20

*P16, L444: "Except for camels, the herds consist of approximately 100 to 200 animals": How many animals did the camel herds consist of?*

**Reply:**

We have revised the sentence and included the number of camels.

**Changes:**

**3.1.1 Bayesian inference method**

. . .

 Except for the camel herd, which consisted of 42 animals, the other herds comprised approximately 100 to 200 animals each .

COMMENT # 1.21

*P18, L484-486: "Observation cases (b) and (c) present an interesting topic for future study: Is it more valuable to have an anemometer on the drone to capture local and temporal wind variations, or to place an anemometer close to the source at a fixed location and use MOST to obtain diffusivity observations?": Could you speculate or form a hypothesis which could be better?*

**Reply:**

We thank the reviewer for their suggestion and fully agree that the experimental design related to the anemometer placement does not have an obviously optimal solution. We have revised the manuscript to clarify the suggested topic for future study and have incorporated a hypothesis.

**Changes:**

**3.1.1 Bayesian inference method**

. . .

 Is it more valuable to equip the drone with an anemometer to capture spatial wind variations, albeit potentially affected by the drone's downwash and motion, or to place a fixed anemometer near the source, which may provide more accurate observations and enable a more reliable application of MOST to derive diffusivity data? We hypothesize that a fixed anemometer may be the superior option within the current framework, which employs an advection-diffusion model based on mean wind conditions. In contrast, a mobile

sensor may prove superior in frameworks that account for spatial variations in the wind field.

COMMENT # 1.22

*P18, L492: Here, you compare the values to IPCC Tier 1 values. You didn't introduce them in the materials section, but you could do it briefly when also introducing the Tier 2 method.*

**Reply:**

We thank the reviewer for their valuable suggestion. As recommended, we have revised the beginning of the IPCC Tier 2 mission values section to introduce the IPCC Tier 1 values as well. We propose the following changes.

**Changes:**

**2.2 IPCC Tier 2 emission values**

. . .

. . .

In addition to drone-based methods, we estimated $CH_4$ emissions from enteric fermentation of ruminant herds using the IPCC Tier 1 and Tier 2 approaches (Paustian et al., 2006). The Tier 1 method uses generalized default values for emission factors $EF$ [kg head$^{-1}$ d$^{-1}$], often specific to regions or continents. In contrast, the Tier 2 method incorporates more detailed, herd-specific or animal-specific data, making Tier 2 values generally more reliable than Tier 1 estimates. This method is based on the typical daily metabolic energy balance of the animals, . . .

COMMENT # 1.23

*P23, L617-619: Were the differences between before and after grazing statistically unsignificant?*

**Reply:**

We thank the reviewer for their insightful question. We have performed a one-tailed Student's t-test to determine if the differences in emissions before and after grazing were statistically significant. We have included the results in the revised version of the manuscript as follows.

**Changes:**

**3.2 Method comparison**

...

Feeding is known to increase $CH_4$ production in ruminants (Amon et al., 2001; Hegarty, 2013).  . Using a one-sided Student's t-test (Student, 1908) on the relative difference in emissions pre- and post-grazing across all animal groups, we found a statistically significant effect of grazing for the Bayesian inference results  (p-value = 0.009), but not for the mass balance results (p-value = 0.194). We observe that seven out of ten  Bayesian inference cases have markedly higher mean emission rates in the afternoon compared to the morning.  For pregnant does and weaner kids,  while the emission estimates increase after grazing,  considerable overlap in the uncertainty ranges makes the  effect ambiguous. In the case of lactating ewes, the difference in emissions before and after grazing is slightly negative  but ambiguous. In contrast, the mass balance results do not consistently demonstrate an increase in $CH_4$ emissions post-grazing, with  substantial increases observed in only  two out of ten cases  : the slick herd and lactating ewes. We consider this to be a promising indicator for the greater reliability and accuracy of our Bayesian inference results compared to our mass balance results.

SMALL CAPS: COMMENT # 1.24

*P23, L623-625: "With the exception of the cows and lactating ewes drone flights, the Bayesian inference estimates pre-grazing are lower than the IPCC Tier 2 value, and the post-grazing estimates are higher than the IPCC Tier 2 value, or the uncertainty ranges of the different methods overlap.": Are the IPCC Tier 2 values some kind of averages, i.e. they can be used to calculate annual emissions? If so, shouldn't you then compare some kind of average of the before and after grazing emission rates with the IPCC values?*

**Reply:**

The IPCC values indeed provide estimated daily emission rates that are based on the typical metabolic energy balance of the animals. As the method accounts for daily energy intake and use, it does not allow for estimates of diurnal variations in emissions. To compare the IPCC values to the drone-based estimates, we converted these daily values to hourly rates. We propose to clarify this in the caption of Fig. 5.

At the start of the Method comparison section, we compare the drone-based results to the IPCC Tier 2 values. As suggested by the reviewer, we propose to include here the average (pre- and post-grazing) results compared to the IPCC Tier 2 results.

To improve clarity and brevity of the manuscript, we suggest removing the text quoted in the comment from the manuscript.

**Changes:**

**Caption of Figure 5**

 . . . , and IPCC emission factors converted from daily to hourly emission rates.

. . .

**3.2 Method comparison**

. . .

In Sec. 3.1.1 and 3.1.2, we  found that the Bayesian inference results for all herds are of the same order of magnitude as the IPCC Tier 1 values.  In contrast, our mass balance results for strong sources (cattle herds) fall within the same order of magnitude as the IPCC Tier 1 values, while those for weak sources (sheep and goat herds) are substantially higher than the IPCC Tier 1 values  in several drone flights. This finding is further supported by comparisons to the herd-specific IPCC Tier 2 values , which are generally regarded as more reliable, though they should not be considered definitive. We compare the average (pre- and post-grazing) emission rate estimates to the IPCC Tier 2 values. We found a relative difference of 16% for the Bayesian inference results and a relative difference of 10% for the mass balance results across flights with strong sources of $Q \approx 700 \ \mathrm{g\,h^{-1}}$ to $Q \approx 1,500 \ \mathrm{g\,h^{-1}}$ (Table B1).  For the flights with weaker sources of $Q \approx 70 \ \mathrm{g\,h^{-1}}$ to $Q \approx 140 \ \mathrm{g\,h^{-1}}$ (Table B1).  , we observed a relative difference of 40% for the Bayesian inference results and a relative difference of 683% for the mass balance results. This disparity suggests that the source term estimation threshold of the Bayesian inference method is  lower than that of the mass balance method applied in our study  Consequently, Bayesian inference may be more effective in estimating weaker sources where the mass balance method  may provide unreliable estimates .

. . .

COMMENT # 1.25

*P23, L633: "In comparison to the Bayesian inference method, the mass balance approach is more straightforward to implement.": Why is that? Do you mean, for example, that the math behind it is simpler/requires less assumptions?*

**Reply:**

We thank the reviewer for their question. We consider the mass balance method to be easier to use due to its lower model complexity, which requires less coding and reduces the need for parameter tuning. We propose to clarify this point in the manuscript as follows.

**Changes:**

**3.2 Method comparison**

. . .

In comparison to the Bayesian inference method, the mass balance approach is more straightforward to  use due to its smaller model complexity, which requires less coding and reduces the need for parameter tuning.

COMMENT # 1.26

*P26, L716-718: "Future applications of the Bayesian framework for source term estimation could extend to diverse natural and anthropogenic sources, such as CH4 emissions from wetlands, hotspots in thawing permafrost, landfills, and wastewater disposal sites.": This paper focused on "point-like" sources but wetlands and landfills are "sparser" and have emissions from a larger area. How well does the method introduced here suit such cases? Are there major issues that need to be addressed before Bayesian interference can be used over such areas?*

**Reply:**

We thank the reviewer for raising this important point. We believe that the Bayesian inference method represents a promising general approach with the potential to map emissions in diverse landscapes. However, as the reviewer pointed out, extending the framework from point-like sources to more sparse or homogeneous sources will require some adjustments to the framework. We propose to revise the manuscript as follows.

**Changes:**

**4.2 Future applications**

. . .

Future applications of the Bayesian framework for source term estimation could extend to diverse natural and anthropogenic sources,  including $CH_4$ emissions from wetlands,  thawing permafrost, landfills, and wastewater disposal sites. Our ultimate aim is to leverage Bayesian inference to comprehensively map these areas, thereby improving our understanding of spatial variations in emissions across diverse landscapes. While the framework is readily applicable to point-like sources, such as thermokarst hotspots, future research should focus on adapting this framework to infer emissions from landscapes characterized by multiple sparse sources that may vary in size and produce overlapping emission plumes. This adaptation may require not only precise estimation of emission rates but also accurate localization of these sources.

COMMENT # 1.27

*Technical corrections*

*P19, L525: Should it be "Figure 5" and not 2?*

**Reply:**

We thank the reviewer for drawing our attention to this point. Both figures demonstrate the intermittency of the emission plume. We have corrected the manuscript accordingly. Please note that the numbering of the figures has changed due to the reorganization of the manuscript.

**Changes:**

**3.1.2 Mass balance method**

. . .

Figures 2 and 4 demonstrates the intermittency of the observed instantaneous plume:

. . .

COMMENT # 1.28

*P19, L536: "negative mission rate" -> "negative emission rate"*

**Reply:**

Fixed.

COMMENT # 1.29

*P23, L625 and L628: "IPPC" -> "IPCC"*

**Reply:**

Fixed.

**REFERENCES**

[revised manuscript text omitted]
 | 118 | 6.2±1.4 | 49±12 | 11.1±2.3▲ | 0.8±0.6• | 0.8±0.6• | 0.9±0.6 | 0.7±0.1 | 79±16 |

---

## Author Response (AR2)

4 June 2025

Dear editor,

Please find the diff-file of the corrected manuscript for publication below.

Kind regards,

Alouette

[revised manuscript text omitted]
 | 118 | 6.2±1.4 | 49±12 | 11.1±2.3▲ | 0.8±0.6● | 0.8±0.6● | 0.9±0.6 | 0.7±0.1 | 79±16 |